# Reconstructing the regulatory programs underlying the phenotypic plasticity of neural cancers

Ida Larsson [1,2,3,5], Felix Held [4,5], Gergana Popova [1], Alper Koc [1], Soumi Kundu [1], Rebecka Jörnsten[4] & Sven Nelander [1] ✉

Nervous system cancers exhibit diverse transcriptional cell states influenced by normal development, injury response, and growth. However, the understanding of these states' regulation and pharmacological relevance remains limited. Here we present "single-cell regulatory-driven clustering" (*scregclust*), a method that reconstructs cellular regulatory programs from extensive collections of single-cell RNA sequencing (scRNA-seq) data from both tumors and developing tissues. The algorithm efficiently divides target genes into modules, predicting key transcription factors and kinases with minimal computational time. Applying this method to adult and childhood brain cancers, we identify critical regulators and suggest interventions that could improve temozolomide treatment in glioblastoma. Additionally, our integrative analysis reveals a meta-module regulated by *SPI1* and *IRF8* linked to an immune-mediated mesenchymal-like state. Finally, *scregclust's* flexibility is demonstrated across 15 tumor types, uncovering both pan-cancer and specific regulators. The algorithm is provided as an easy-to-use R package that facilitates the exploration of regulatory programs underlying cell plasticity.

Nervous system cancers in adults and children share a common trait: tumor cells exist in multiple transcriptional states, partly resembling the diversification of cells during normal embryonic development. In adult glioblastoma (GBM), cells resemble neural progenitor cells (NPCs), oligodendrocyte progenitor cells (OPCs), or astrocytes (ACs)[1]. Additionally, a substantial fraction of cells display a mesenchymal (MES)/injury response-like profile linked to monocyte populations[2–6]. In childhood cancers such as diffuse midline glioma (DMG), tumor cells transition from a proliferative, OPC-like state to more differentiated states resembling either AC- or oligodendrocyte (OC)-like lineages[7]. In medulloblastoma (MB), the four main tumor subgroups differ in the proportion of undifferentiated versus differentiated tumor cells and their lineage resemblance[8].

There are strong reasons to believe that the cell states found in a tumor have implications for disease progression. For instance, invasive GBM cells mimic the migration mechanisms of neuronal cells[9,10], whereas tumor initiation is mediated by quiescent stem-like cells[11]. Further, it has been reported that recurrent therapy-resistant GBM tumors display a higher fraction of MES cells[12], possibly due to a phenotypic shift from non-MES to MES[13]. However, despite their role in disease progression, recurrence and drug resistance, our knowledge about the mechanisms regulating cell states is limited. Specifically, to understand transcriptional regulation, it is essential to identify sets of transcription factors (TFs) whose activity directly impacts a specific state phenotype. Moreover, to identify drug targeting opportunities, it is important to pinpoint sets of druggable proteins, particularly kinases, linked to these states.

The increasing availability of scRNA-seq data repositories presents new opportunities to uncover such regulation with high precision. However, existing data analysis methods are not developed with

[1]Department of Immunology, Genetics and Pathology, Uppsala University, SE-751 85 Uppsala, Sweden. [2]Department of Pediatric Oncology, Dana-Farber Cancer Institute Boston, MA, USA. [3]Broad Institute of MIT and Harvard, Cambridge, MA, USA. [4]Mathematical Sciences, Chalmers University of Technology, SE-412 96 Gothenburg, Sweden. [5]These authors contributed equally: Ida Larsson, Felix Held. ✉e-mail: sven.nelander@igp.uu.se

this goal in mind. Frequently used clustering methods, including graph-based/community detection approaches[14,15], k-means and hierarchical clustering, do not contain regulatory predictions; they focus on grouping cells with similar gene expression profiles. Similarly, methods specifically developed for clustering genes based on scRNA-seq data lack regulatory predictions[16–18]. In the bulk RNA-seq setting, a standard method to identify regulators of gene signatures involved first estimating a gene regulatory network (GRN), followed by post-processing to identify regulators linked to signature genes[19,20]. Transferring this strategy to the scRNA-seq setting, many methods to create GRNs exist, both adapted from methods applied to bulk RNA-seq data[21,22] or specifically developed for scRNA-seq data[23]. However, GRNs created from scRNA-seq data are less reliable with individual links being hard to reproduce[24,25] and with a lower retrieval of known functional links compared to bulk RNA-seq data[26]. Adding additional layers of data, primarily ATACseq or CHIPseq measurements[27–29], can alleviate the problem but comes with its own limitations. Existing methods are not fast enough to process millions of cells, and restricting the model to only TFs makes it less amenable to modeling the impact of pharmacologically relevant gene classes, such as kinases. Thus, the development of fast and accurate methods to reconstruct gene regulatory programs from scRNA-seq data is important, if not essential.

Here, we describe a method for the fast construction of regulatory programs, composed of sets of target genes whose variation is well explained by regulators selected by the algorithm. The method is well-suited for large data sets and detecting critical regulators of cell states. Applied to scRNA-seq data, it jointly detects modules of co-expressed target genes and regulators, such as transcription factors and kinases, linked to each module. Compared to methods of similar scope, it is fast, robust, and flexible. As proof-of-principle, we demonstrate the applicability of our algorithm through four use cases. First, we apply it to a data set from peripheral blood mononuclear cells (PBMC) and benchmark it against the most comparable algorithm we can find, SCENIC+[30]. Second, we apply it to scRNA-seq data previously generated by our group[31] to predict regulatory interventions to potentiate temozolomide (TMZ) treatment in GBM. Third, we integrate 13 data sets from the developing brain and nervous system cancers to investigate the regulatory landscape of neuro-oncology. We find a meta-module regulated by the transcription factors *SPI1* and *IRF8*, which shows strong resemblance to the previously described immune cell-induced MES-like state. Finally, we perform a pan-cancer analysis, integrating data sets from 15 tumor types to define pan-cancer and cancer-specific regulators. The algorithm is available as an easy-to-use R package.

## Results

### A regulatory-driven clustering method

The key goal of our analysis is to identify regulators of cell states. To achieve this, we present an approach to detect regulatory programs from scRNA-seq data alone, based on a mathematical framework that directly models the interaction between regulators and responding target gene sets (Fig. 1a). Note that, in this context, a regulatory program is not a detailed network linking individual genes, e.g.[16,21–23], but rather a robust, higher-level model where regulators influence groups of target genes. These regulators can be selected based on known regulatory functions, such as TFs or kinases, or can be chosen for their potential in experimental follow-up. The target gene groups, referred to as modules, represent broader cellular programs or states, like astrocyte-like differentiation or the cell cycle.

Our model assumes that genes belong to either of two categories: tentative regulators or tentative targets. Given a volume of data $\mathbf{Z}$, organized as cells × genes, we split the data in two parts along genes, a regulatory part $\mathbf{Z}_r$, and a target part $\mathbf{Z}_t$. For technical reasons described in "Methods", it is required that the number of processed cells is at

least double that of the number of regulators. Both $\mathbf{Z}_r$ and $\mathbf{Z}_t$ are further randomly split along cells into two sets of training and assessment cells. We then perform a clustering task by identifying groups of genes (modules) in $\mathbf{Z}_t$ that correspond to sets of co-regulated genes, and for each such module, we perform a regulatory program reconstruction task by identifying a small number of genes in $\mathbf{Z}_r$ that are the likely regulators. These steps are repeated until configurations stabilize.

To fit regulatory programs from data computationally, we developed an alternating two-step scheme iterating between determining the most predictive regulators (regulatory program reconstruction) for each of a pre-specified number $K$ of modules and using these optimal regulators to allocate target genes into $K$ modules (clustering).

Step 1 is approached using cooperative-Lasso[coop-Lasso][32], on the training set. In this step, we search for a sparse set of regulators that can be linked to most genes in a module, each regulator with the same sign (positive vs. negative). The sign consistency is an important feature of the model, as it is highly unlikely that a regulator has different modes of interaction with target genes belonging to the same module. A sparsity penalty parameter controls the number of regulators assigned to each module. To solve the optimization problem efficiently, we apply over-relaxed Alternating Direction Method of Multipliers[ADMM][33,34] to minimize the coop-Lasso objective function.

Once regulators have been assigned to modules, we move on to Step 2, where the task is to refine the target gene modules. We approach this by re-estimating coefficients per module using sign-constrained non-negative least squares[NNLS][35] on the training set and an observation-based allocation scheme using the assessment set. Target genes whose predictive $R^2$ is below a threshold across all modules are marked as noise and placed in a rag-bag cluster. This ensures that outliers do not overly distort the clustering result. The structure of our algorithm is shown in Box 1 and algorithmic details can be found in "Methods".

There are several optional inputs that can guide the algorithm. Of note is the possibility of providing prior information on the relationship between genes in the form of a gene × gene matrix, where a non-zero entry indicates that there is a known link between the genes. A known link can, for example, be that both genes appear in the same biological process. Also, as a feature of the method, users who wish to run Step 1 alone on pre-defined gene sets can do so.

### Comparison between *scregclust* and SCENIC+ shows significantly overlapping results

Here, we demonstrate our entire workflow on a publicly available dataset from peripheral blood mononuclear cells (PBMC). Starting from a matrix of raw gene counts, the data was normalized and formatted (see details in "Methods"). As previously mentioned, potential regulators can be all TFs, all kinases, or any custom regulator annotation of interest. For a direct comparison with the newly released SCENIC+[30], a method inferring cell state-specific enhancer-driven gene regulatory networks from scRNA- and scATAC-seq data, we used all TFs as potential regulators. We did not provide our algorithm with any prior information. Once the algorithm converged, it detected a network of nine gene modules and 53 TFs interacting with one or several of these gene modules.

The PBMC cells clustered into 8 clusters, and using immune cell type marker expression, we annotated each cluster and defined gene signatures of each immune cell type (Fig. 1b). To functionally annotate the gene modules defined by *scregclust*, we overlapped these with the gene signatures of the immune cell types and quantified the overlap using the Jaccard Index (Fig. 1c). Our algorithm successfully distinguishes modules upregulated in CD8+ T cells, CD4+ T cells, B cells, NK cells, FCGR3A+ monocytes, CD14+ monocytes, and plasmacytoid dendritic cells (pDCs). Among the strongest regulators, we find many examples of known regulators, e.g., *PAX5* (B cells), *TBX21* (NK cells), *CUX2* (dendritic cells), and *LEF1* (T cells), to mention a few (Fig. 1c). As a

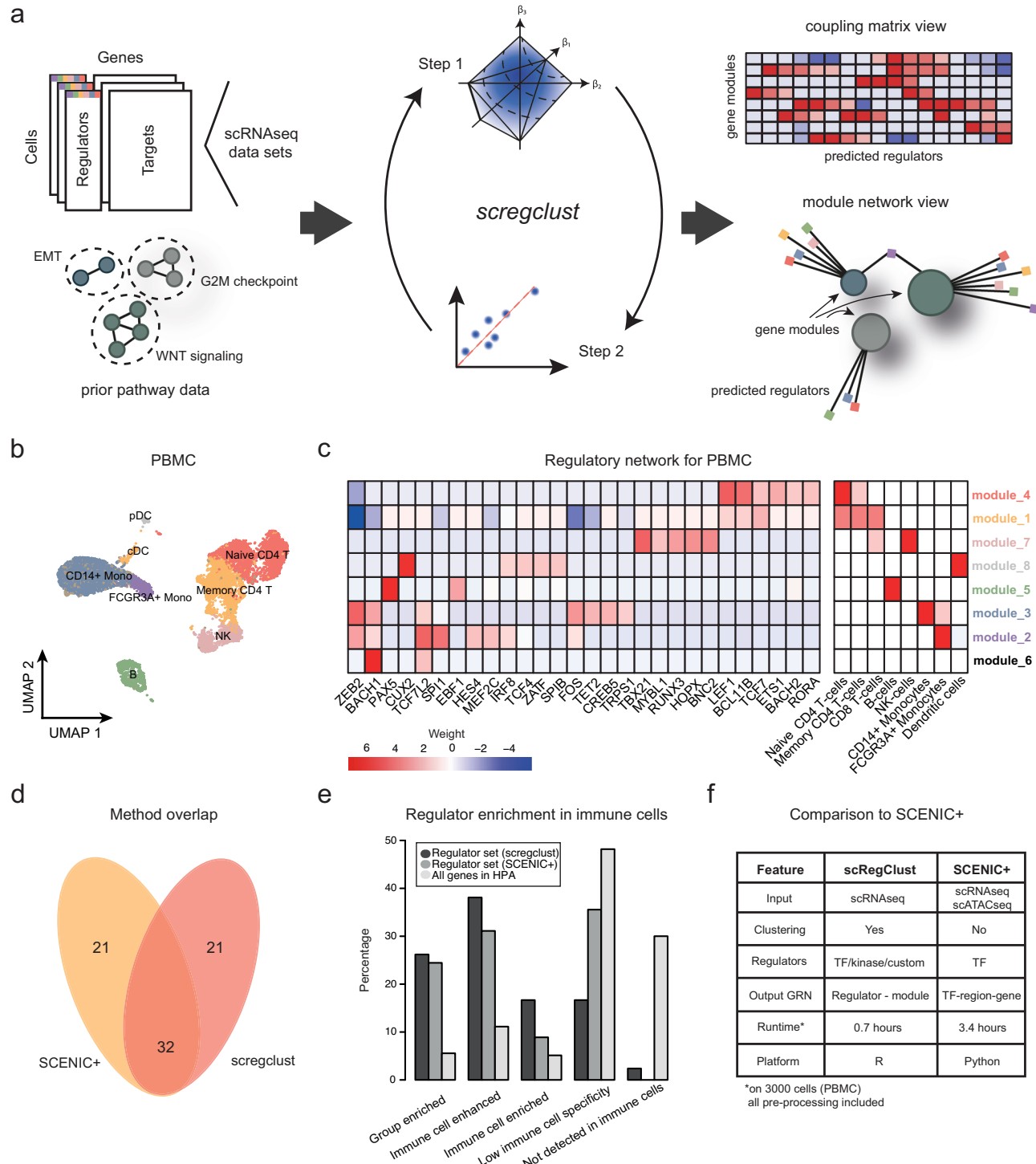

**Fig. 1 | A regulatory-driven clustering method. a** A schematic overview of the algorithm pipeline from gene expression data to regulatory networks. **b** UMAP embedding of the PBMC cells colored according to immune cell type assignment. Abbreviations: Mono monocytes, T T-cells, B B-cells, NK natural killer cells, pDC plasmacytoid dendritic cells, cDC classical dendritic cells. **c** The regulatory table inferred by *scregclust*. In the left panel, rows correspond to modules and columns to regulators (TFs). The scale in the heatmap goes from strong positive regulation (red) to strong negative regulation (blue), as indicated by the key. In the right panel, rows correspond to modules and columns to immune cell type gene signatures. Heatmap color indicates degree of overlap between module gene content and gene signatures, as quantified by Jaccard Index. Module name color (rows) correspond to colors in (**b**), indicating the most enriched immune cell type. Source data are provided as a Source Data file. **d** Venn diagram showing the overlap in identified regulators between SCENIC+ and *scregclust*. **e** Annotation of immune cell specificity of the identified regulators by *scregclust* and SCENIC+ according to the Human Protein Atlas (HPA)[36]. Immune enriched/enhanced indicate that a gene's mRNA level is at least 4-fold higher in an immune cell type compared to all other tissues, as defined by the authors of the HPA. The regulator sets are contrasted to the average annotation of all genes in the HPA (*n* = 13157). **f** Comparison between *scregclust* and SCENIC+.

## BOX 1

# High-level overview of the *scregclust* algorithm

**Input**: Pre-processed expression data for target genes ($\mathbf{Z}_t$) and regulators ($\mathbf{Z}_r$). Optionally, an initial clustering ($\mathbf{\Pi}$) and an indicator matrix ($\mathbf{J}$) describing prior knowledge.

**Initialization**: Split data randomly into two sets (($\mathbf{Z}_{t,1}$, $\mathbf{Z}_{r,1}$)) and (($\mathbf{Z}_{t,2}$, $\mathbf{Z}_{r,2}$)), and, if not provided in input, find initial clustering ($\mathbf{\Pi}$) by applying k-means++ to the cross-correlation matrix of ($\mathbf{Z}_{t,1}$) and ($\mathbf{Z}_{r,1}$).

**for** cycle ($c$) from (1) to maximum cycle **do**
 Store current clustering ($\mathbf{\Pi}$)
 **for** each cluster ($i$) **do** ▷ Step 1: Selection
 *Given the current clustering* ($\mathbf{\Pi}$):
 • Identify active regulators ($R_i$) and determine their signs ($\mathbf{s}_i$).
 **end for**

 **for** each cluster ($i$) **do** ▷ Step 2a: Re-estimation
 *Given active regulators* ($R_i$) *and their signs* ($\mathbf{s}_i$):
 • Re-estimate coefficients for each target gene.
 • Compute normalized likelihood ($\mathbf{L}_j^{(i,\,l)}$) for each gene ($j$) and
 observation ($l$).
 • Incorporate prior information if supplied.
 **end do**

 **for** each target gene ($j$) **do** ▷ Step 2b: Allocation
 • Compute ($R_{i,j}^2$) for each cluster ($i$) and perform one of the
 following.
 (a) Sort out noisy genes into the rag-bag cluster if ($R_{i,j}^2$) across all
 clusters is too low.
 (b) Update the cluster allocation of ($j$) according to a majority
 vote across observations.
 **end do**

 **if** ($\mathbf{\Pi}$) is equal to *any* previously stored
 clustering **then** ▷ Stopping criteria
 Stop iteration.
 **end if**

**end for**

point of comparison, SCENIC+ also identified 53 (activator) TFs, out of which 32 overlap with our identified regulators (OR = 30.37, $p < 10^{-16}$, Fisher's exact test; Fig. 1d). To gain a more detailed understanding of our output, we examined the functional annotation in the Human Protein Atlas[36] for the predicted regulators, with a particular focus on the method-specific (non-overlapping) regulators. For the *scregclust*-predicted regulators that had an annotation in the HPA, 81% are classified as being immune cell-enriched/enhanced compared to 64% of regulators predicted by SCENIC+ and a baseline 22% for all genes in the atlas (Fig. 1e). For the non-overlapping genes, 81% of the *scregclust*-specific regulators are enhanced or enriched in immune cells and have an immune cell type annotation that agrees with the one predicted by *scregclust*. For the SCENIC+-specific regulators, the same number is 43% (Supplementary Data 1).

Jointly, the comparison between *scregclust* and SCENIC+ on PBMC data showed a strong (OR = 30.37) association between the two methods' results, even though *scregclust* uses scRNA-seq data alone. In contrast to SCENIC+, which relies on pre-defined cell types for each cell in the primary data, *scregclust* performs clustering of target genes into modules, which can then be functionally annotated to match cell types. As previously mentioned, SCENIC+ is restricted to identifying TFs as regulators, due to its reliance on scATAC-seq data, while *scregclust* in principle can consider any category of genes as potential regulators. The speed of the two algorithms is also remarkably different, with *scregclust* being three times faster than SCENIC+ (Fig. 1f).

**Regulatory-driven clustering: performance and robustness**
To further assess our algorithm, we performed a range of tests on both simulated count data and on real-life datasets.

First, to demonstrate that penalty parameters in the *scregclust* package can be selected through a data-driven approach, yet still achieve stable clustering and selection performance, we fitted *scregclust* to simulated data and evaluated its performance using True and False Positive Rates (ROC curve) for correct links between regulators and targets, and the Adjusted Rand Index (ARI) and cluster homogeneity for clustering accuracy (see "Methods"). The simulated count data (negative binomial distributed) mimics the real data distribution, but with a known module and regulator structure (see Supplementary Information for details). We introduced two measures to guide the choice of the sparsity penalty parameter: Predictive $R^2$, which assesses how well the identified regulators predict each module, and Regulator Importance, which measures the change in Predictive $R^2$ when a single regulator is omitted.

Our simulation study demonstrates how *scregclust* can recover regulatory programs in a tunable fashion (Fig. 2a) and that Predictive $R^2$ and Regulator Importance can guide the selection of appropriate penalty parameters (Fig. 2b). Increasing the penalty parameter reduces the number of active regulators, gradually decreasing Predictive $R^2$ until it becomes too strong and important regulators are excluded. Conversely, Regulator Importance increases as fewer regulators are included until it becomes inflated. Identifying the 'elbow' point of these trends helps set the penalty, and we recommend testing at least five values to find this point. Another key parameter is the initial number of modules $K$. Unlike k-means, our algorithm can produce empty modules, enforced by specifying a minimum non-empty cluster size. Simulations showed that *scregclust* tends to combine ground truth modules with regulatory overlap when $K$ is underspecified and splits them when $K$ is overspecified. We use silhouette scores to guide $K$ selection (Fig. 2C), which measure how well a target gene fits within its module compared to the nearest module. A good choice of $K$ results in high silhouette scores for all genes allocated to a module. If $K$ is too large, some modules are split which results in negative silhouette scores for these modules. High average silhouette scores and Predictive $R^2$ per module indicate a good clustering (Fig. 2D) and choice of $K$. We recommend starting with an initial guess for $K$ and testing a range of values to find the optimal number of modules. In this simulation, Fig. 2B,D show that an optimal penalty parameter is around 0.05 to 0.1 and an initial number of modules $K = 7$ leads to a good trade-off between silhouette scores, number of resulting modules and $R^2$.

Next, we benchmarked various aspects of *scregclust* against well-established, state-of-the-art methods for gene clustering and GRN reconstruction (see Supplementary Information for details). Comparisons against traditional gene clustering algorithms (k-means and hierarchical clustering) as well as methods specifically developed for scRNA-seq data (Celda[17]) demonstrate a highly stable behavior of *scregclust* when clustering genes (Supplementary Fig. 1). While clustering is an important aspect, the primary functionality of *scregclust* is the joint clustering of target genes and sign-consistent identification of

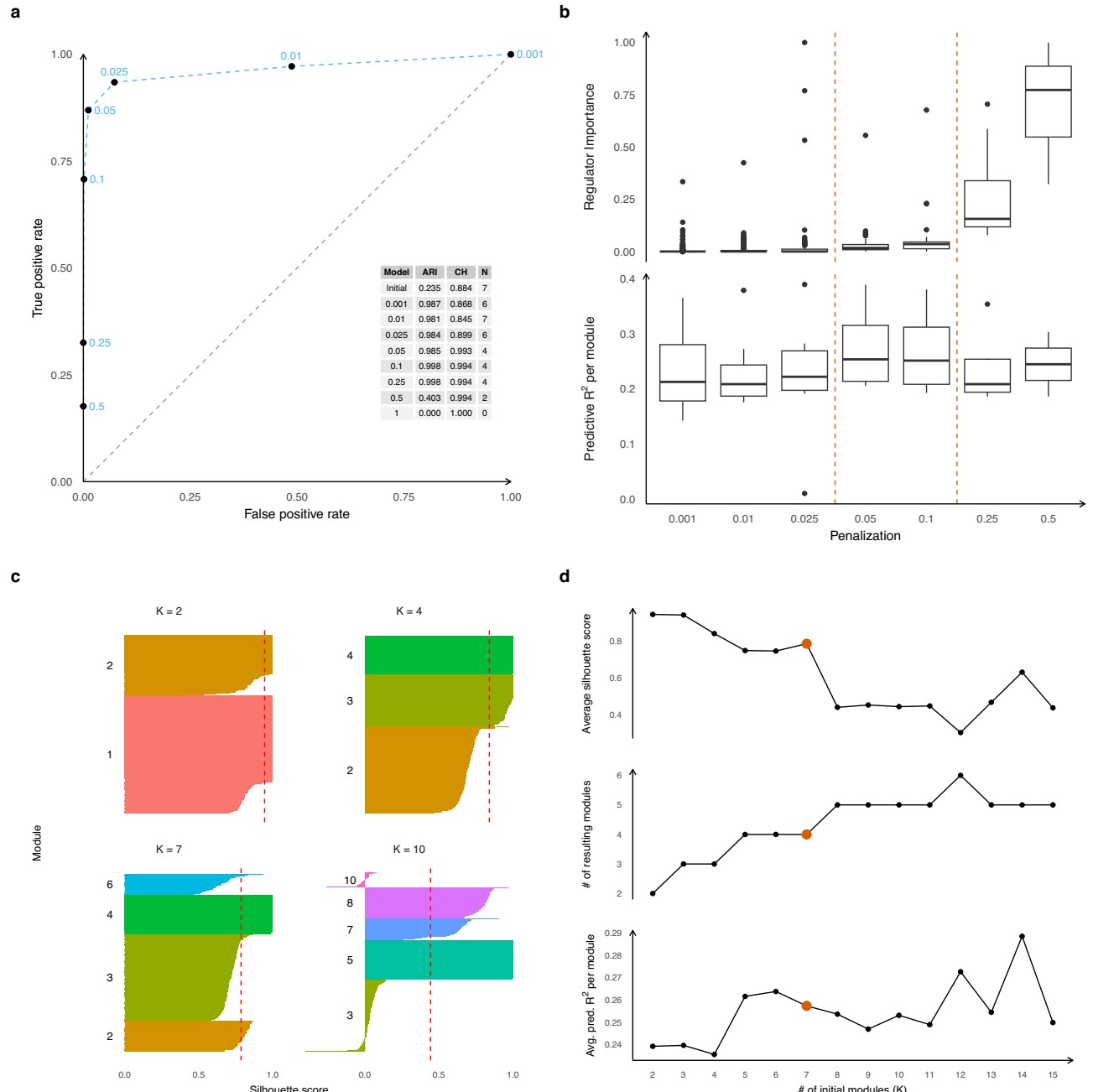

**Fig. 2 | Method tuning and analysis tools.** Results from *scregclust* run on simulated scRNAseq count data (see the Supplementary Material for details on data generation). **a** Average true positive rate (TPR) shown against average false positive rate (FPR) in a ROC curve (dashed blue line), illustrating the quality of groundtruth regulator identification. TPR and FPR are computed per target gene and then averaged ($n = 268, 270, 271, 271, 271, 80$ from smallest to largest penalization). The corresponding penalty parameter is shown with each average. A table of adjusted Rand indices (ARI) between the estimated clustering and the true clustering, the average cluster homogeneity (CH), and the resulting number of modules (N) is shown as an inset. **b** Boxplots of predictive $R^2$ per non-empty module ($n = 6, 7, 6, 4, 4, 4, 2$, cfr. N in Part A) and importance per regulator associated with at least one non-empty module ($n = 900, 453, 96, 39, 25, 11, 3$) shown across a

progression of seven penalty parameters. Dashed lines indicate a region of solutions that demonstrates our selection rule. Boxplots consist of center lines (median), box bounds (1st and 3rd quartile), and upper and lower whiskers. Upper whiskers are drawn from the upper box bound to the largest data point but no further than 1.5 times the inter-quartile range (IRQ), analogous for lower whiskers. All data points not covered by box and whiskers are shown as dots. **c** Silhouette scores for each module for runs different initial K. Dashed red lines indicate the average silhouette score. Target genes have been grouped by module and sorted by decreasing silhouette score. Colors and labels to the left of each group indicate the module. **d** Average silhoutte scores, resulting number of modules, and average predictive $R^2$ as a function of the initial number of modules K. Optimal selection indicated by red points.

regulators. To evaluate this, we also assessed *scregclust* for its ability to establish robust, sign-consistent associations between gene clusters and regulators, comparing it with methods such as PPCOR[37], WGCNA[38], and the scRNA version hdWGCNA[16], combined with community detection, GRNboost2[22] combined with k-means and PIDC[23] combined

with Celda (Supplementary Figs. 2–7). Notably, *scregclust* uniquely ensures sign consistency. Additionally, *scregclust* demonstrates better stability in regulator identification compared to WGCNA, hdWGCNA and PPCOR. GRNboost2, due to its design, lacks robustness in identifying cluster-consistent sets of regulators.

Overall, our evaluation of the performance of *scregclust* highlights its proficiency in clustering accuracy, robustness and uncovering regulatory interactions across diverse data scenarios.

### Targeting regulators of an OPC-like state potentiates temozolomide treatment

Encouraged by our benchmarking results, we applied *scregclust* to complex scRNA-seq data sets from nervous system cancers to explore two key questions. First, could the algorithm predict interventions, such as drugs or gene perturbations, to push GBM cells into a temozolomide (TMZ)-sensitive state? Second, could it integrate scRNA-seq data across nervous system cancers and the developing brain to identify meta-modules with shared regulation and function?

One of the questions raised during our previous work on cell state transitions in GBM was how to effectively target an entire cell state and not just individual marker genes of these states. The question was prompted by our prediction that, in the primary cell culture U3065MG, the minimal intervention needed to potentiate TMZ treatment is to block transitions to what we termed "state 5", a state with an OPC-like, invasive profile[31]. To demonstrate how *scregclust* integrates into a typical scRNA-seq analysis workflow, we addressed how to modify state transitions. Specifically, we applied *scregclust* to scRNA-seq data from U3065MG generated in[31] using either TFs or kinases as potential regulators (Fig. 3a). Without prior information on target gene module assignment, *scregclust* clustered the genes into 9 (TF) or 6 (kinase) modules, predicting 60 regulators of these modules (Fig. 3b). Modules were characterized by their similarity to known gene signatures using the Jaccard index.

We found that *scregclust* defines a group of modules corresponding to state 2, a highly proliferative progenitor state. By studying the functional profile of these modules further (Fig. 3b), it was evident that several modules were related to proliferation and the cell cycle. Two of the modules represent the earlier phases of cell cycle (G1, G1S, S) while remaining modules represent the later phases (S, G2M, M). Several regulators identified for these modules agree with the module profiles, e.g. *E2F1* and *TK1* regulating the earlier phases of cell cycle[39,40] while *AURKA* and *CENPA* are mostly active during G2M/M-phase[41,42]. Another interesting observation is the astrocyte-like state 4, which is predicted to be positively regulated by *ID3*, known to induce astrocyte differentiation through the BMP pathway signaling[43].

To address our initial question on how to suppress state 5 and potentiate TMZ treatment, we focused on the regulation of modules corresponding to this state (Fig. 3c). The top kinase regulator was *PDGFRA*, consistent with state 5's OPC-like profile, as *PDGFRA* is a known regulator of OPCs in normal development[44] and frequently amplified in the OPC-like GBM state[1]. Other strong kinase regulators included *DDR1*, expressed in oligodendrocytes during brain development[45], and *ERBB3*, also implicated in oligodendrocyte lineage development and the OPC-like state in GBM[46]. The top TF regulators were *SOX6* (positive regulation) and *YBX1* (negative regulation). *SOX6* belongs to the group D family of SOX TFs, which regulate several stages of oligodendrocyte development. The combined predictions from[31] and *scregclust* suggested that blocking *PDGFRA*, *DDR1*, *ERBB3*, or *SOX6*, or increasing *YBX1* activity, would potentiate TMZ treatment. Experimental validation using CRISPR/Cas9-mediated knockdown of *DDR1* or *SOX6* in U3065MG cells confirmed that both knockdowns enhanced TMZ response, significantly reducing cell viability at higher drug doses (Fig. 3d).

We also tested combining TMZ with the tyrosine-kinase inhibitor (TKI) dasatinib (Fig. 3e), targeting *PDGFRA*, *ERBB3*, and *DDR1*. While simultaneous treatment showed no synergistic effect, pre-treating cells with dasatinib followed by combined treatment resulted in a significant synergistic effect (BSS > 10) (Fig. 3f). This suggests that pre-treatment with dasatinib depletes state 5 cells, making the tumor more susceptible to TMZ. A known predictor of TMZ response is

methylation of the *MGMT* gene[47,48], and from previous work, we know that U3065MG cells are *MGMT* methylated[49]. To test whether our combination treatment results were related to *MGMT* methylation status, we included additional primary cell cultures with varying *MGMT* methylation status in the experiment (Supplementary Fig. 9A). We observed a trend where *MGMT* methylated cell cultures had a higher synergy score than *MGMT* unmethylated cell cultures, but the difference was not statistically significant (Supplementary Fig. 9B). These results indicate that *MGMT* methylation likely contributes to the treatment effect in terms of cells being more responsive to TMZ, but it is not the main explanatory factor for the identified synergistic combination.

Building on these results, we broadened our investigation's scope to explore regulators of cell plasticity in nervous system cancers. To that end, we included more GBM single-cell data sets, as well as data from other cancers of the nervous system and corresponding healthy tissue.

### The regulatory landscape of neuro-oncology

We used *scregclust* to integrate the regulatory programs from the developing brain[50–55], normal adrenal gland[56], GBM[1,50,57–59], MB[8,53,60] and neuroblastoma (NB)[56]. We ran the algorithm for each data set individually and merged the resulting regulatory tables (Fig. 4a). In the merged regulatory table (Fig. 4b), the rows correspond to regulators (TFs) and the columns to gene modules derived from each individual study. The top annotation bars indicate the disease type and study that each module originate from. It is evident that certain meta-modules emerge (Fig. 5), consisting of several modules with similar gene content from different studies that are regulated by the same regulators.

Two meta-modules, 1 and 7, represented actively proliferating cells from all types (normal tissue, GBM, MB and NB). Meta-module 1 was primarily driven by positive regulation by *CENPA* and *HMGB2*, while meta-module 7 was more strongly regulated by *HMGB1* and *DEK*. Our model suggests a subdivision of MB and normal samples along a gradient regulated by *HMGB1/2* and *DEK* vs *NEUROD1*, where *NEUROD1* promotes non-proliferating neuronal-like NPC2-like cells (meta-module 3), and *HMGB1/2* and *DEK* promotes neural progenitors with active proliferation.

Two meta-modules, 2 and 6, were enriched for hallmark signatures of hypoxia and apoptosis, as well as the Ivy atlas signature pseudopalisading cells around necrosis (CT pan), probably representing a general stress response in the cells. These were regulated by *YBX1* (#2) and members of the Fos and Jun family of TFs (#6). An astrocyte-like meta-module (#4) in normal and MB cells was linked to *ID3*, *ID4* and *HOPX*. In GBM, these cells were clustered in meta-module 2 due to their additional enrichment for stress response signatures. An OPC-like meta-module (#13) was identified that was driven by e.g. *OLIG1*, *MYRF* and *SOX10*.

There were two distinct mesenchymal meta-modules, 5 and 15. One mesenchymal meta-module (#5) coincided with an enrichment of microglial signatures and was primarily driven by *SPI1* and *IRF8* in normal, GBM and MB cells. An explanation for this meta-module is that it captures the phenotype of cells undergoing a mesenchymal shift induced by tumor-associated macrophages (TAMs) or microglia, an occurrence that has been thoroughly described in the context of GBM biology in previous studies[3–6]. The enrichment of microglia-related gene signatures in disease modules is probably due to this previous interaction between tumor and immune cells. *SPI1* and *IRF8* could therefore be candidate regulators of the observed mesenchymal shift in GBM cells. *SPI1* is a pioneer TF, i.e. one that can bind directly to condensed chromatin and thereafter recruit other, non-pioneer TFs, to the site. The known normal function of *SPI1* is in controlling the cell fate decision of hematopoietic cells, e.g. in macrophage differentiation. After binding to regulatory elements in the genome, it regulates gene expression by recruiting other

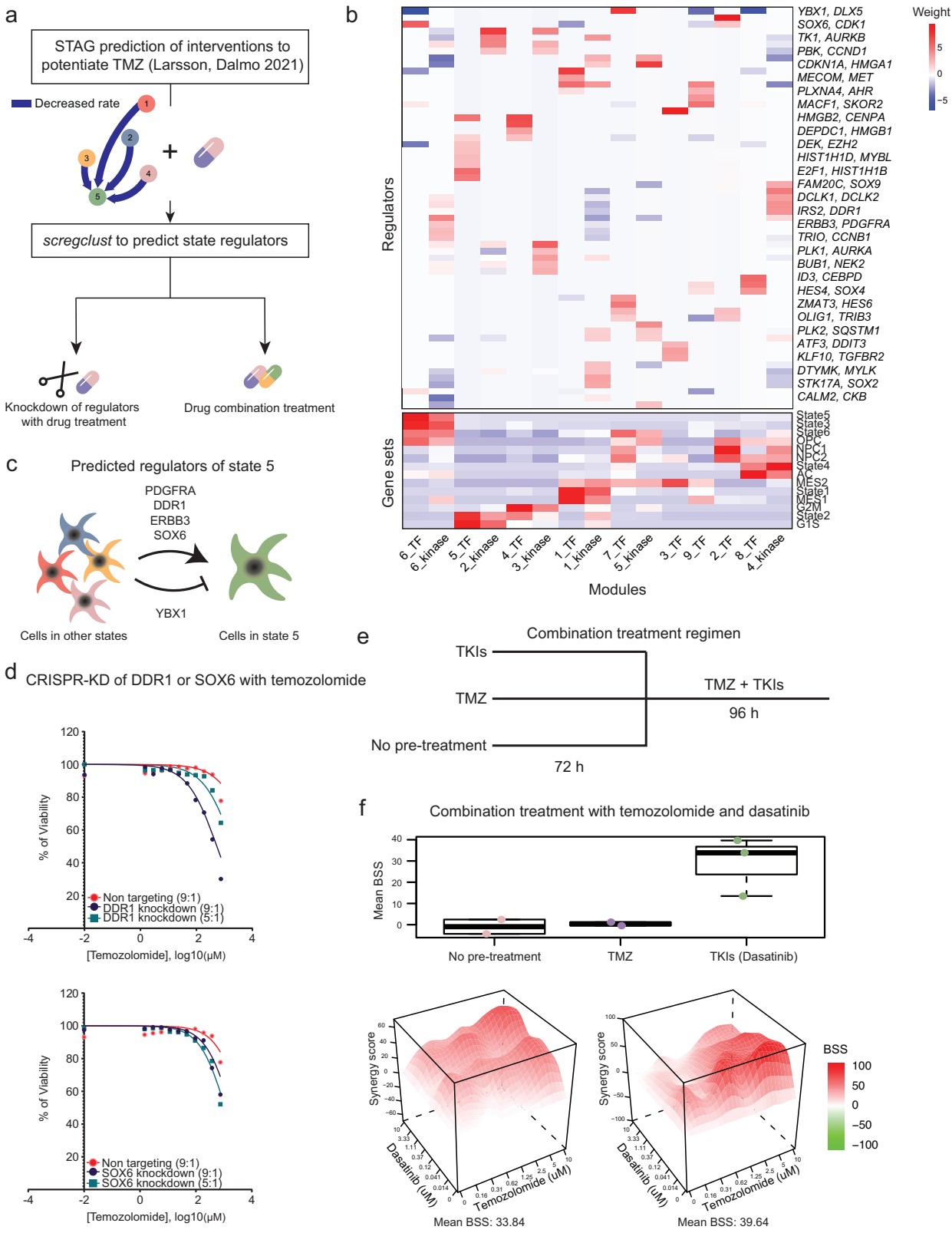

transcription factors, such as interferon regulatory factors, e.g. *IRF8*[61,62]. *IRF8* has been previously identified as a candidate gene involved in the immune evasion of GBM cells. Gangoso et al. (2021) observed that *IRF8* was upregulated in GSC cells in an immuno-competent mouse model following immune attack, and since *IRF8* is normally a myeloid-specific master transcription factor, they termed this immune evasion strategy "myeloid-mimicry".

The other mesenchymal metamodule (#15) showed no enrich-ment of microglial signatures, but instead showed enrichment for the Ivy atlas signature microvascular proliferation. This meta-module was interesting in the sense that it had regulators that depended on disease category. GBM samples were regulated by *FOXS1* and *HEY1*, while MB samples were regulated by *AEBP1* and *TBX18*. Normal tissue samples were driven by previously mentioned regulators but also *MEF2C* and

**Fig. 3 | *scregclust* predicts a way of potentiating temozolomide treatment in U3065MG cells. a** Flowchart of analysis. In our previous paper[31], we predicted that in the primary cell line U3065MG, TMZ treatment would be potentiated by an intervention that blocked transitions to "state 5". Using *scregclust*, we predict regulators of state 5 and follow up on these predictions by combining TMZ with either CRISPR/Cas9-mediated knockdown of state 5 regulators, or combination treatment with a drug inhibiting these regulators. **b** The merged regulatory table from *scregclust*, with gene modules as columns and regulators as rows. Bottom panel display similarity of each module to gene signatures from ref. 31 and ref. 1, as defined by Jaccard index. For ease of reading, two regulator (gene) names are indicated per row, the gene name after the comma are for the row below. Source data are provided as a Source Data file. **c** Schematic of the predicted regulation of state 5. *PDGFRA*, *DDR1*, *ERBB3* and *SOX6* positively regulated state 5 and should be knocked down to block transitions to state 5. *YBX1* negatively regulated transitions

to state 5 and should be overexpressed to block transitions to state 5. **d** Dose response curves for TMZ-treated cells with CRISPR/Cas9-mediated knockdown of DDR1 (top) and SOX6 (bottom). 10 doses of TMZ were tested, in a range from 750 μM to 1.25 μM. For each dose, duplicate measurements were taken. **e** Schematic of the three combination treatment arms. Cells were either pre-treated with tyrosine kinase inhibitors (TKIs) or TMZ for 72 h, or no pre-treatment, followed by combination treatment with TMZ and TKIs for 96 h. **f** Boxplot showing the median bliss synergy score (BSS) for the three treatment arms (biological replicates, $n = 2$, 2, and 3 in the order displayed in the figure), all replicates (top), and synergy landscape for the two combination experiments where the highest synergy scores were obtained (bottom). Boxplot for TKIs consist of center lines (median), box bounds (1st and 3rd quartile), and upper and lower whiskers. Upper whiskers are drawn from the upper box bound to the largest data point but no further than 1.5 times the inter-quartile range (IRQ), analogous for lower whiskers.

---

*TSC22D4*. A third meta-module (#12) showed a slightly weaker enrichment for the microvascular proliferation signature and included all types. This was driven by several TFs from the ERG family (*ERG*, *FLI*, *ETS1*) and *SOX17*, to name a few. Several of these regulator interactions are consistent with their described role in literature, e.g. *FLI1* which is a prognostic marker in astrocytoma[63] and is predicted to be a regulator of perivascular-like glioma cells[63] and the endothelial potential of myogenic progenitors[64]. Similarly, the role of *SOX17* as a promotor of tumor angiogenesis has been described in mice[65].

Our model also suggests a subdivision of NB samples (meta-modules 9 and 14) along an adrenal-to-mesenchymal axis, driven by *ZNF90*, *MEIS2* and *MYC*, each of which was positively linked to mesenchymal-like signatures and negatively linked to ADRN-like signatures. Remaining meta-modules (#8, 10, 11) were specific to one type and driven by just one or two regulators.

Similarly to the TF regulatory landscape, the kinase map (Supplementary Fig. 10) revealed several likely links and made interesting predictions. For instance, along the oligodendrocytic lineage, *AATK* is linked to a meta-module in both GBM and normal developing brain that is enriched for oligodendrocyte differentiation. *PDGFRA* and *DGKB* is linked to a meta-module with an OPC-like profile, whereas *PDGFRB* is linked to invasiveness and microvascular proliferation signatures. *CKB* regulates a meta-module of NPC signature targets in normal developing brain, GBM and MB, and suppresses cell cycle genes in MB and GBM. *CCND1* suppresses NPC markers. Mature astrocytes were primarily linked to *NTRK2* activity. *CCL2* drives a microglia-like signature in both normal developing brain and MB. As in the TF map, known cell cycle driving kinases converge on similar modules in GBM, MB and normal developing brain, e.g. *AURKA*, *PLK1* and *CCNB2*. Still there was an MB-specific *TK1*-driven cell cycle module. *AK4* links to a GBM-specific module enriched for cellular tumor genes. We also found some hits of unclear significance, like *SGK1* regulating a common meta-module in GBM, MB and normal developing brain.

## A pan-cancer study of regulatory programs reveals regulators of intratumoral heterogeneity

Finally, we extended above described analysis to include more cancer types than nervous system cancers. Specifically, we ran *scregclust* on scRNA-seq data from patient-derived samples of acute myeloid leukemia (AML), breast cancer, colorectal cancer, head and neck cancer (HNSCC), liver cancer, non-small-cell lung cancer (NSCLC), osteosarcoma, ovarian cancer, pancreatic cancer, prostate cancer, renal cancer and skin cancer (Table 1). In addition, we included one each of previous runs on GBM, MB and NB. As before, *scregclust* was run on each data set separately, and the resulting regulatory tables were merged into a pan-cancer regulatory landscape with regulators as rows and individual target gene modules as columns (Fig. 6a). By annotating gene modules according to both cancer type and broader cancer

category (brain/spinal cord, carcinoma, leukemia, sarcoma), we could search for regulators that were specific for certain cancer types. Of the total 650 identified regulators, 122 were specific for one type of cancer. Glioblastoma was the cancer type with the most cancer-specific regulators, followed by ovarian, liver and renal cancer (Fig. 6b).

To characterize the target gene modules, we compared them with gene signatures derived from ref. 66, where 41 meta-programs (MPs) were defined, representing the hallmarks of intratumoral heterogeneity across cancer types (Supplementary Fig. 11). To highlight a few examples; (i) we find a small cluster of target modules that enriches for MP 24 (Cilia) and are regulated by *FOXJ1*, a known master regulator of ciliogenesis[67], (ii) we find that the strongest positive regulator of MP 12 (EMT I) is *PRRX1*, a TF known to be involved in metastasis through EMT in various cancers[68], and (iii) we find one target gene module derived from the NSCLC dataset that strongly enriches for MP 31 (alveolar) and is regulated by *NKX2-1*, a TF that in normal lung development specifies alveolar cell identity and has been found to be a prognostic marker in NSCLC[69,70] (Supplementary Data 2).

Taken together, our combined analyses demonstrates the power of *scregclust* as a convenient tool for investigating regulators of intratumoral heterogeneity and generating biological hypotheses that can be tested experimentally.

## Discussion
The high extent of intratumoral heterogeneity in nervous system cancers and the plastic behavior of tumor cells are major obstacles in the search for more efficient therapies against these often deadly diseases. The enormous amount of scRNA-seq data that have been generated to map intratumoral heterogeneity presents great opportunities to understand the regulation of transcriptional cell states and cell state transitions, but it also challenges us to develop suitable methods to properly analyze the data. In this work, we have addressed the lack of computational methods to identify such regulators of intratumoral heterogeneity and present a method that simultaneously detects gene modules and their regulators from scRNA-seq data. The utility of our method is demonstrated through several use cases, where it is applied to real data, and through a thorough investigation of model properties using synthetic data.

By associating a regulator model with each module, *scregclust* improves substantially on initial clustering results derived from the correlation of target genes and regulators. We further develop scores, such as $R^2$ per module, regulator importance, and a silhouette score, to help the user with parameter choices and to support the interpretation of the resulting regulatory network. In our first use case, we apply *scregclust* to the PBMC data set and compare our results to those of SCENIC+. We show that we can, in an unsupervised fashion, recapitulate signatures of the known cell types present in the data as well as several known regulators of each cell type. The results show a strong association to that of SCENIC+, but obtained in shorter time and with only one

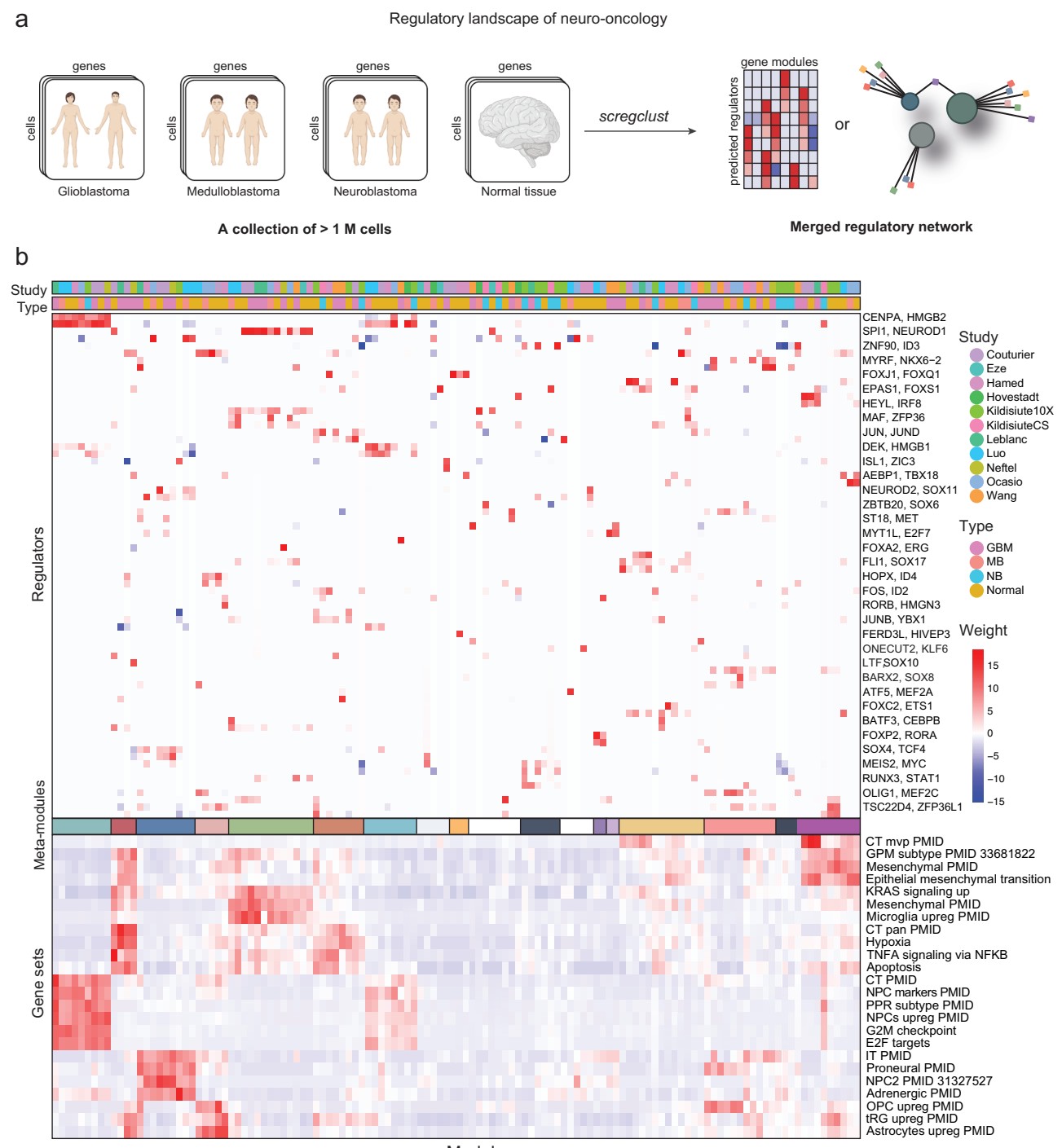

**Fig. 4 | The regulatory landscape of neuro-oncology. a** Schematic overview of the analysis. Created in BioRender. Nelander, S. (2023) BioRender.com/b96y619. **b** Middle panel is the regulatory table from *screglust*, with modules as columns and regulators (TFs) as rows. Top panel are annotation bars indicating what type and study each module originate from. Meta-modules (Fig. 5) are indicated by color in the bar below the middle panel. For ease of reading, two regulator (gene) names are indicated per row, the gene name after the comma are for the row below. Bottom panel display similarity of each module (Jaccard index) to a database of neuro-oncology related gene sets, derived from studies indicated by PubMed ID (PMID). Abbreviations: CT cellular tumor, MVP microvascular proliferation, GPM glycolytic/plurimetabolic, pan pseudopalisading cells around necrosis, NPC neural progenitor cells, PPR proliferating progenitor cells, IT infiltrative tumor, OPC oligodendrocyte progenitor cells, tRG truncated radial glia. Source data are provided as a Source Data file.

data source. We do however emphasize that although the methods produce a similar output, they differ in several important technical aspects, and should be treated as complementing rather than competing methods. Similarly, in our benchmarking we demonstrate the stability and reliability of *screglust* in comparison to state-of-the-art clustering and network construction algorithms. However, users should

note that the aim of *screglust* is not to reconstruct full gene-to-gene regulatory networks and therefore users will benefit from selecting tools based on their specific goal, whether that is to focus on regulatory programs with *screglust* or reconstructing full gene-to-gene networks.

In our second use case, we aim to demonstrate how *screglust* integrates into a typical scRNA-seq analysis workflow and how it can be

## The regulatory meta-modules

| | Meta-module | Interpretation | Regulators | Normal | GBM | MB | NB |
|---|---|---|---|---|---|---|---|
| | 1 | Cycling cells | *HMGB1/2+, CENPA+, DEK+* | ■ | ■ | ■ | ■ |
| | 2 | Necrosis, hypoxia, apoptosis | *YBX1-, ID3+* | ■ | ■ | □ | □ |
| | 3 | Non-proliferating NPC2-like cells | *HMGB1/2-, NEUROD1/2+, DEK+, SOX11+, SOX4+, TCF4+* | ■ | ■ | ■ | ■ |
| | 4 | Astrocyte-like cells | *ID3+, ID4+, HOPX+, ZFP36L1+, MEF2C+* | ■ | □ | ■ | ■ |
| | 5 | Microglial-like mesenchymal cells | *SPI1+, IRF8+, MAF+, ZFP36+, NME2+* | ■ | ■ | ■ | ■ |
| | 6 | Stress response | *FOXJ1+, JUN+, JUND+, FOS+, JUNB+, HMGN3+, NR4A1+* | ■ | ■ | □ | ■ |
| | 7 | Cycling cells | *HMGB1/2+, CENPA+/-, DEK+, NEUROD1-* | ■ | ■ | ■ | ■ |
| | 8 | FOXJ1-driven | *FOXJ1+* | □ | ■ | □ | ■ |
| | 9 | NB non-adrenergic cells | *ZNF90+, MEIS2+, MYC+, STAT1+* | ■ | ■ | □ | ■ |
| | 10 | FOXP2, RORA-driven infiltrative tumor | *FOXP2+, RORA+* | ■ | □ | ■ | ■ |
| | 11 | MYT1L-driven infiltrative tumor | *MYT1L+* | □ | ■ | □ | ■ |
| | 12 | SOX17-driven mvp | *ERG+, FLI+, ETS1+, EPAS1+, FOXQ1+, SOX17+, ZFP36+, ID3+* | ■ | ■ | ■ | ■ |
| | 13 | OPC-like cells | *OLIG1+, MYRF+, SOX10+, TSC22D4+, NKX6-2+(/-)SOX6+* | ■ | ■ | ■ | ■ |
| | 14 | NB adrenergic cells | *ZNF90-, MEIS2-, MYC-* | ■ | ■ | □ | ■ |
| | 15 | Mvp-signature mesenchymal cells | *FOXS1+, HEY1+, AEBP1+, TBX18+, NME2+, MEF2C+, TSC22D4+* | ■ | ■ | ■ | □ |

**Fig. 5 | The regulatory meta-modules.** Description of the meta-modules defined in Fig. 4. The colors to the left correspond to the middle color bar in Fig. 4. The regulators listed regulate at least two of the individual modules included in the meta-module and all diseases represented in the meta-module are indicated by a filled box.

used to derive hypotheses that can be tested experimentally. We apply our algorithm to a previously generated scRNA-seq datasets from GBM cells and identify several regulators of an OPC-like, invasive GBM state and combine TMZ and dasatinib, a tyrosine kinase inhibitor targeting the state regulators, to potentiate TMZ treatment. Dasatinib has in previous studies shown anti-migratory effects[71], consistent with state 5 having an invasive profile, and were tested in combination with TMZ in a clinical trial as treatment against GBM[72]. However, this trial was terminated before proceeding to phase II. We find that the combination treatment is only synergistic in our U3065MG cell line when we first pre-treat the cells with dasatinib, followed by combined treatment with both drugs, and speculate whether this means that state 5 first needs to be depleted for the combination treatment to be synergistic. These speculations need to be more carefully investigated by measuring whether the protein levels of state 5 markers decrease following dasatinib treatment.

In our third and fourth use cases, *scregclust* is applied to data sets from nervous system cancers (GBM, MB and NB) and the developing brain, as well as to data from 15 tumor types in a pan-cancer analysis, and used to derive insights about the diseases that warrant further investigation. First, by performing an integrative study of the regulatory programs of GBM, MB, NB and the developing brain, we find that *SPI1* and *IRF8* are candidate regulators of the mesenchymal shift induced by TAMs and/or microglia in GBM. These regulators can be an important piece of the puzzle to understand the mechanism behind the observed mesenchymal transition, which appears to be a strategy for the tumor cells to evade the immune system and become increasingly resistant to treatment. Second, in the pan-cancer analysis we demonstrate the broad applicability of our method across cancer types and define both cancer-specific and pan-cancer regulators.

For future developments, we see several possible extensions of the algorithm. At present, *scregclust* has the functionality to include prior knowledge of gene-gene relationships to guide the clustering of target genes. It would however be interesting to explore the possibility

of including prior knowledge in the regulator-target gene relationship as well, e.g. by favouring connections that have support in an external data set (ATAC-seq, CHIP-seq etc). In addition to this, we see extensions related to non-additive contributions to the regulation of a module, extended regulation modeling to allow for feedback regulation, or automatic partitioning in regulator- and target genes.

To conclude, we present an algorithm that operates on scRNA-seq data alone to construct regulatory programs consisting of regulators and target gene modules, and use it to understand the regulatory mechanisms of cell state plasticity. The algorithm is provided as an easy-to-use R-package and can be applied to any scRNA-seq data set.

## Methods

### Ethics statement
Primary glioblastoma cell lines were used in this study, as detailed below in section "Knockdown experiments" and "Drug combination treatments". These cell lines were obtained from the human glioblastoma cell culture (HGCC) resource[49]. For the establishment of the HGCC resource, tumor sample collection was approved by the Uppsala Regional Ethical Review Board number 2007/353. (As described previously, patient signed informed consent was obtained for the HGCC collection, and no monetary compensation was offered[49]).

### Datasets
All publicly available data sets used in this publication are listed in Table 1, including the data repository from where they were downloaded. The notation "3CA" refers to the Curated Cancer Cell Atlas provided by the Tirosh Lab[66].

### Data processing and formatting
If available, raw counts were downloaded from the indicated source in Table 1. The data was processed before further analysis in R using the Seurat package[73] (v5.1.0). Cells containing <200 genes and genes present in <3 cells were filtered out. Data generated from a UMI-based

**Table 1 | Summary of data sets used in this publication**

| Study | Data repository | Disease |
|---|---|---|
| Bi et al., 2021 | 3CA/kidney | Renal cell carcinoma |
| Caron et al., 2020 | 3CA/hematologic | Acute lymphoblastic leukemia |
| Chen et al., 2021 | 3CA/prostate | Prostate cancer |
| Couturier et al., 2020 | 3CA/brain | Developing brain |
| Couturier et al., 2020 | 3CA/brain | Glioblastoma |
| Darmanis et al., 2017 | 3CA/brain | Glioblastoma |
| Eze et al., 2021 | UCSC Cell Browser | Developing brain |
| Hamed et al., 2022 | GEO: GSE200202 | Developing brain |
| Hovestadt et al., 2019 | 3CA/brain | Medulloblastoma |
| Hwang et al., 2022 | 3CA/pancreas | Pancreatic ductal adenocarcinoma |
| Ji et al., 2020 | 3CA/skin | Squamous cell carcinoma |
| Kildisiute et al., 2021 | Neuroblastoma Cell Atlas | Neuroblastoma |
| Kildisiute et al., 2021 | Neuroblastoma Cell Atlas | Normal fetal adrenal gland |
| Kürten et al., 2021 | 3CA/head and neck | Head and neck squamous cell carcinoma |
| Larsson, Dalmo et al., 2021 | ArrayExpress E-MTAB-9296 | Glioblastoma |
| Leblanc et al., 2022 | GEO: GSE173278, GSE193884 | Glioblastoma |
| Luo et al., 2021 | Short Read Archive: GSE156633 | Developing cerebellar granule |
| Luo et al., 2021 | Short Read Archive: GSE156633 | Medulloblastoma |
| Ma et al., 2019 | 3CA/liver and biliary | Hepatocellular carcinoma |
| Manno et al., 2021 | Sequence Read Archive: PRJNA637987 | Developing brain |
| Neftel et al., 2019 | 3CA/brain | Glioblastoma |
| Ocasio et al., 2019 | GEO: GSE129730 | Medulloblastoma |
| Ocasio et al., 2019 | GEO: GSE129730 | Cerebellum |
| Pelka et al., 2021 | 3CA/colorectal | Colorectal cancer |
| Qian et al., 2020 | 3CA/lung | Lung cancer |
| Qian et al., 2020 | 3CA/ovarian | Ovarian cancer |
| Rendeiro et al., 2020 | 3CA/hematologic | Chronic lymphocytic leukemia |
| Riether et al., 2020 | 3CA/hematologic | Acute myeloid leukemia |
| Wang et al., 2019 | 3CA/brain | Glioblastoma |
| Weng et al., 2019 | GEO: GSE122871 | Developing brain |
| Wu et al., 2020 | 3CA/hematologic | Acute myeloid leukemia |
| Wu et al., 2021 | 3CA/breast | Breast cancer |
| Zhou et al., 2021 | 3CA/sarcoma | Osteosarcoma |
| PBMC | 10X website | Human blood cells |

single-cell sequencing protocol (10X Chromium, CEL-seq2) were normalized using sctransform[74,75] (v0.4.1), while expression levels for remaining data sets were quantified using $\log_2(\text{TPM}/10 + 1)$, as suggested by Tirosh et al. (2016). When needed, non-malignant cells were filtered out from the data set based on the provided metadata from the authors or the 3CA-portal[66].

To run *scregclust*, the input data needs to be packaged according to a given format. In addition to the gene expression matrix (p x n), an indicator vector (p x 1) needs to be provided. In the vector, a 1 indicates

that the gene is a potential regulator. For the case of the input data being human and either transcription factors or kinases should be considered as potential regulators, we provide functionality (the function scregclust_format) to format the normalized data matrix to conform with the desired input of *scregclust*. For any other scenario (species other than human, a custom regulator list) we leave it to the user to format their input according to the specified format. Key parameter values without a default value chosen for each dataset in the analysis are shown in Supplementary Data 3.

## Downstream analysis

For the analyses presented in Figs. 4 and 5 *scregclust* was run on each dataset separately. Each run generates a "regulatory table", a matrix of dimension regulator x modules. Thereafter, the regulatory tables from each run were combined by merging the matrices by row names. Any NAs introduced due to non-overlapping regulators were replaced by 0, and only rows and columns with an absolute sum > 0 were kept. Finally, the merged regulatory table was centered and scaled and the generated z-scores were used for plotting.

The enrichment analyses in Figs. 1, 3, and 4, as well as Supplementary Figs. 6 and 7, were performed by comparing the overlap between the genes in each target module and gene signatures of known cell types or biological processes. The overlap was quantified using Jaccard Index.

The assessment of the overlap between the regulator set from *scregclust* and SCENIC+ (Fig. 1D) was done using the function fisher.test() in R. The background population was defined as all transcription factors in the human genome ($n = 1820$), since this is all genes that could potentially be nominated by either of the algorithms as regulators.

## Clustering algorithm

We developed a two-step alternating algorithm to simultaneously perform clustering on a set of target genes as well as to associate a set of regulators with each cluster (module). In a first step, given an initial clustering of the data, the algorithm determines the linearly most predictive regulators for each cluster as well as whether the regulator acts stimulating or repressing. Then, in a second step, these optimal regulators are used to allocate target genes into clusters. Prior knowledge about target gene relationships can be included to guide cluster allocation. These two steps are repeated until configurations stabilize or a maximum number of steps is reached. The algorithm was implemented as an R-package and can be found on GitHub at https://github.com/scmethods/scregclust.

**Statistical model.** We consider a sign-constrained linear regression model for target gene expression $z_t$ specific to each of the $K$ clusters. Let $\mathbf{z}_r$ be regulator expression, a vector of length $p_r$. $R_i \subset \{1, ..., p_r\}$ contains the indices of regulators selected for cluster $i$, $\mathbf{s}_i \in \{-1, 1\}^{|R_i|}$ contains regulator signs, $\boldsymbol{\beta}_i$ is a vector of length $|R_i|$ of non-negative regression coefficients, and $\sigma_i^2 > 0$ is a variance parameter for cluster $i$. Let $\pi_i \in \{0, 1\}$ with $\sum_{i=1}^{K} \pi_i = 1$ be cluster labels such that target gene expression $z_t$ follows the model

$$z_t | \pi_i = 1 \sim N(\mathbf{z}_r^{(R_i)^\top} \text{diag}(\mathbf{s}_i)\boldsymbol{\beta}_i, \sigma_i^2). \tag{1}$$

Extending the model to a vector of responses $\mathbf{z}_t$ containing gene expression for all $p_t$ target genes, we assume that the correlation between target genes is primarily captured by their regulators. Therefore, $\mathbf{z}_t$ are assumed conditionally uncorrelated within and between clusters. Components $\mathbf{z}_t^{(j)}$ are then modeled according to equation (1) with cluster labels $\pi_{i,j}$, coefficients $\boldsymbol{\beta}_{i,j}$, and variance $\sigma_{i,j}^2$. Selected regulators $R_i$ and regulator signs $\mathbf{s}_i$ are cluster-specific and remain unchanged. Coefficients are collected in $|R_i| \times p_t$ matrices $\mathbf{B}_i = (\boldsymbol{\beta}_{i,1}, \dots, \boldsymbol{\beta}_{i,p_t})$ and cluster labels $\boldsymbol{\Pi}$ are collected in a $K \times p_t$ matrix

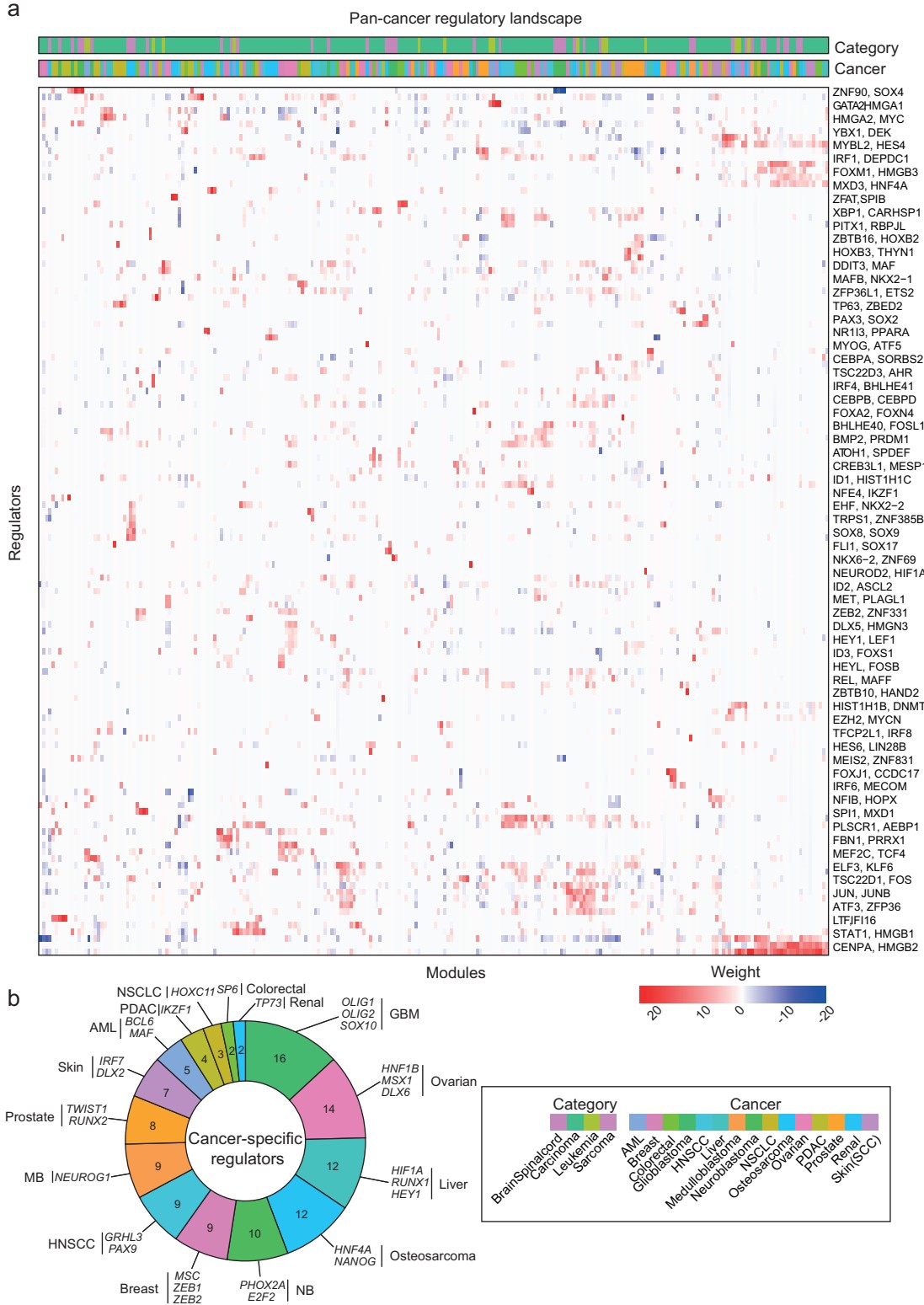

**Fig. 6 | The pan-cancer regulatory landscape. a** Merged regulatory table from *scregclust*, with modules as columns and regulators (TFs) as rows. Top panel are annotation bars indicating what cancer type and cancer category each module originate from. For ease of reading, two regulator (gene) names are indicated per row, the gene name after the comma are for the row below. Source data are provided as a Source Data file. **b** The number of cancer-specific regulators per cancer type, colored as in (**a**). A couple of specific regulators per cancer type are displayed. Glioblastoma had the most cancer-specific regulators (16), with *OLIG1*, *OLIG2* and *SOX10* being three examples.

**Table 2 | Overview of symbols used in the description of the _scregclust_ algorithm**

| Symbol | Description |
|---|---|
| Problem setup: | |
| $n$ | Total number of cells in the input matrix |
| $p_t$ | Number of target genes |
| $p_r$ | Number of regulators |
| $\mathbf{Z}_t, \mathbf{Z}_r$ | target gene and regulator expression ($n \times p_t$ resp. $n \times p_r$ matrices) |
| $K$ | Desired number of clusters |
| $\mathbf{\Pi}$ | $K \times p_t$ cluster membership matrix for target genes with $\mathbf{\Pi}^{(i,j)} \in \{0, 1\}$ and $\sum_{i=1}^{K} \mathbf{\Pi}^{(i,j)} \leq 1$ |
| $\mathbf{J}$ | $p_t \times p_t$ indicator matrix describing prior knowledge of biological relationships between target genes (e.g. pathway co-occurence) |
| For data splits $d = 1, 2$: | |
| $n_d$ | Number of cells in the $d$-th data split |
| $\mathbf{Z}_{t,d}$ | target gene expression used in the $d$-th data split ($n_d \times p_t$ matrix) |
| $\mathbf{Z}_{r,d}$ | regulator expression used in the $d$-th data split ($n_d \times p_r$ matrix) |
| For each cluster $i = 1, ..., K$: | |
| $C_i$ | Set of target genes in cluster $i$ |
| $R_i$ | Set of regulators associated with cluster $i$ |
| $N_i$ | Maximum number of regulators associated with cluster $i$ |
| $\mathbf{s}_i$ | Sign vectors containing one sign for each regulator in $R_i$ |
| $\mathbf{B}_i$ | non-negative regression coefficients ($|R_i| \times p_t$ matrix) |
| $\sigma_{i,j}^2$ | positive variance parameter for each target gene and cluster |
| Optimization-related ($i = 1, ..., K, j = 1, ..., p_t$): | |
| $\lambda$ | Positive penalty parameter used in coop-Lasso |
| $\mathbf{w}_i$ | Positive weight vector of length $p_r$ for each cluster $i$ in coop-Lasso |
| $\mathbf{B}_{i,\text{OLS}}$ | Ordinary least squares estimates of the regression coefficients in cluster $i$ ($p_r \times |C_i|$ matrix) |
| $\mathbf{B}_{i,\text{CL}}$ | Coop-lasso estimates of the regression coefficients in cluster $i$ ($p_r \times |C_i|$ matrix) |
| $\tau$ | Non-negative threshold for rag-bag clustering |
| $p_{i,j}$ | Prior probability of target gene $j$ being in cluster $i$ |
| $\mathbf{L}_j$ | $K \times n_2$ likelihood matrix for target gene $j$ |
| $\mathbf{v}_j$ | Vector of $n_2$ votes for the cluster assignment of target gene $j$ |
| $\mu$ | Prior strength in [0, 1] for trade-off between likelihood and prior in cluster allocation |
| Validation measures ($i = 1, ..., K, j = 1, ..., p_t, k = 1, ..., p_r$): | |
| $R_i^2$ | Predictive $R^2$ for cluster $i$ |
| $R_{i,-k}^2$ | Predictive $R^2$ for cluster $i$ with regulator $k$ omitted |
| $I_{i,k}$ | Importance of regulator $k$ in cluster $i$ |
| $R_{i,j}^2$ | Predictive $R^2$ for target gene $j$ predicted by regulators in $R_i$ |
| $S_j$ | Silhouette score for target gene $j$ |
| Output: | |
| $\mathbf{T}$ | Regulatory table providing a summary of regulator strength in each cluster ($p_r \times K$ matrix) |

such that $\mathbf{\Pi}^{(i,j)} = \pi_{i,j}$. Given $n$ independent observations in $n \times p_t$ matrix $\mathbf{Z}_t$ and $n \times p_r$ matrix $\mathbf{Z}_r$, the log-likelihood function for the data then becomes

$$
\begin{aligned}
\log p(\mathbf{Z}_t | \mathbf{Z}_r, \mathbf{\Pi}) &= \sum_{l=1}^{n} \sum_{i=1}^{K} \sum_{j=1}^{p_t} \mathbf{\Pi}^{(i,j)} \log N(\mathbf{Z}_t^{(l,j)} | \mathbf{Z}_r^{(l,R_i)} \text{diag}(\mathbf{s}_i) \mathbf{B}_i^{(:,j)}, \sigma_{i,j}^2) \\
&= \sum_{i=1}^{K} \sum_{j=1}^{p_t} \mathbf{\Pi}^{(i,j)} \log N(\mathbf{Y}^{(:,j)} | \mathbf{Z}_r^{(:,R_i)} \text{diag}(\mathbf{s}_i) \mathbf{B}_i^{(:,j)}, \sigma_{i,j}^2 \mathbf{I}).
\end{aligned}
\tag{2}
$$

Each cluster is therefore described by a selection of regulators $R_i$ and their signs $\mathbf{s}_i$ encoding the assumption that a regulator acts stimulating

(positive sign) or repressing (negative sign) on all target genes in the cluster. Coefficient values and residual variances are considered to be target gene-specific.

We do not allow regulators and targets to switch roles. Rather it is assumed that the assignment of genes to be tentative regulators or targets is provided and fixed at the start of the algorithm. _scregclust_ aims to quantify the regulatory impact of a pre-selected set of regulators and does not aim to estimate a complete gene regulatory network.

## Algorithmic outline of _scregclust_

The algorithm takes as its input two matrices $\mathbf{Z}_t$ and $\mathbf{Z}_r$ containing $n$ rows of matching observations, typically cells, on $p_t$ target genes and $p_r$ tentative regulator genes in their columns, respectively. In addition, the algorithm requires the number $K$ of desired clusters and an initial clustering of the target genes into $K$ clusters. If the latter is not supplied, an initial clustering is produced as described below under "Preprocessing and initialization".

Table 2 contains an overview of all symbols used in the description of the _scregclust_ algorithm.

To cluster target genes and to select regulators associated with each cluster we start by considering to minimize the negative of the log-likelihood function presented in equation (2)

$$
\arg\min_{\mathbf{\Pi}, R, \mathbf{B}, \mathbf{s}, \sigma} \frac{1}{2} \sum_{i=1}^{K} \sum_{j=1}^{p_t} \mathbf{\Pi}^{(i,j)} \left( n \log(\sigma_{i,j}^2) + \frac{1}{\sigma_{i,j}^2} \left\| \mathbf{Z}_t^{(:,j)} - \mathbf{Z}_r^{(:,R_i)} \text{diag}(\mathbf{s}_i) \mathbf{B}_i^{(:,j)} \right\|_F^2 \right)
$$

such that $|R_i| \leq N_i$, $\mathbf{s}_i^{(k)} \in \{-1, 1\}$, $\mathbf{B}_i \geq \mathbf{0}$ and $\sigma_{i,j} > 0$

for all $i = 1, ..., K, j = 1, ..., p_t, k = 1, ..., p_r$.

$$\tag{3}$$

The size constraint $|R_i| \leq N_i$ ensures that only a subset of regulator is associated with each cluster. Note, that we do allow regulators to appear in multiple clusters.

The direct solution of this optimization problem, however, requires to solve a combinatorial number of regression problems, which is computationally too demanding in practice. Instead, we used a data-driven approximation to solve equation (3).

## Data-driven approximation

We proceed in the fashion of alternating clustering algorithms, such as k-means, by alternating between (1) determining the structure of clusters ($R_i$ and $\mathbf{s}_i$), and (2) re-assigning cluster membership ($\mathbf{\Pi}$).

**Preprocessing and initialization.** Observations of target genes and regulators (or a randomly reduced subset) are randomly split into two sets to allow for unbiased estimation of the model quality during cluster allocation below. The ratio is user-defined but 50-50 by default. If the observations are stratified, the sample assignment of each observation can be supplied and splitting is performed within each sub-group. We write $\mathbf{Z}_{t,d}$ and $\mathbf{Z}_{r,d}$ for the $d$-th data split containing $n_d$ observations ($d = 1, 2$). The first data split is regarded as a training set, whereas the second data split functions as an assessment set. We therefore compute the preprocessing below on the training set and apply the same values to the assessment set.

During data splitting, the user can choose whether or not to center the target genes within each sub-group defined by the sample stratification. In addition, after splitting, both target and regulatory genes are centered individually (without regard to stratification) and regulatory genes are scaled to standard deviation 1.

If no initial cluster membership is provided, the cross-correlation matrix of $\mathbf{Z}_{t,1}$ and $\mathbf{Z}_{r,1}$ is computed. The k-means++ algorithm[76] with the cross-correlation matrix as an input and multiple restarts is then used to find an initial clustering of the target genes into $K$ clusters. Note that this step takes regulators into account and is therefore not equivalent to the clustering of only target genes.

**Determining a regulatory model for each cluster.** Given the cluster membership of target genes, the goal of Step 1 is to determine which regulators are linearly most predictive for the target genes in each cluster. If cluster membership is considered to be known, the optimization problem in equation (3) can be solved separately for each cluster. However, due to the computational complexity of the optimization problem, selection of at most $N_{i,r}$ out of $p_t$ regulators as well as their signs, it is not possible to solve it exactly. In our scenario for cluster $i$, there are $\sum_{k=0}^{N_{i,r}} \binom{p_t}{k} \cdot 2^k$ candidate sets of regulators and their signs to consider.

The size constraint $|R_i| \leq N_{i,r}$ in (3) is equivalent to replacing $\mathbf{Z}_r^{(:, R_i)}$ with $\mathbf{Z}_r$, considering $\mathbf{B}_i$ to be a $p_r \times p_t$ matrix, and introducing $\ell_0$ regularization on rows of $\mathbf{B}_i$. This approach ensures that at most $N_{i,r}$ rows of $\mathbf{B}_i$ are non-zero and the indices of these rows correspond to $R_i$. A common approach to ensure computability of $\ell_0$ penalized problems is relaxation of the constraint to a convex norm instead. Here, groups of coefficients are considered that are either zero or non-zero simultaneously. This is a typical scenario covered by the group-Lasso[77]. In addition, to ensure sign-consistency of the selected regulators, an extension of the group-Lasso known as the cooperative-Lasso[coop-Lasso,[32]] is used to solve the regulator selection problem.

Applied to our case, the coop-Lasso solves the following optimization problem

$$\mathbf{B}_{i,\mathrm{CL}} = \arg\min_{\mathbf{B}} \left[ \frac{1}{2} \sum_{j \in C_i} \frac{1}{n_1 \sigma_{i,j}^2} \left\| \mathbf{Z}_{t,1}^{(:,j)} - \mathbf{Z}_{r,1}\mathbf{B}^{(:,j)} \right\|_F^2 + \lambda \sum_{k=1}^{p_r} \mathbf{w}_i^{(k)} \left( \| \mathbf{B}_+^{(k,:)} \|_2 + \| \mathbf{B}_-^{(k,:)} \|_2 \right) \right], \tag{4}$$

where $C_i$ is the set of all target genes in cluster $i$, $\lambda$ is a penalty parameter related to $N_{i,r}$ above, controlling the amount of regulators that will be selected, $\mathbf{w}_i$ is a vector of weights that can be cluster-specific and will be described below, $\mathbf{B}^{(k,:)}$ refers to the $k$-th row in $\mathbf{B}$, and $\mathbf{B}_+^{(k,:)}$ as well as $\mathbf{B}_-^{(k,:)}$ are defined by setting all negative or all positive elements in $\mathbf{B}^{(k,:)}$ to zero, respectively.

The coop-Lasso selects which regulators, i.e. which rows of $\mathbf{B}$, are included in the model by setting the coefficients of the deselected groups to zero. In addition, the coop-Lasso aims for sign-coherence in each group and induces sparsity within groups, deselecting target genes not affected by some of the regulators. This means that typically the rows of the estimated coefficient matrix $\mathbf{B}_{i,\mathrm{CL}}$ will have all positive or all negative sign.

To determine the sign of each regulator, we assign $\mathbf{s}_i^{(k)} = \mathrm{sgn}\left( \sum_j \mathbf{B}_{i,\mathrm{CL}}^{(k,j)} / |C_i| \right)$. The set of active regulators $R_i$ is equal to the indices of non-zero rows in $\mathbf{B}_{i,\mathrm{CL}}$. This way, the coop-Lasso in equation (4) provides an approximation to the optimization problem in equation (3) for fixed cluster membership.

To make computations on a matrix of coefficients efficient, we apply over-relaxed Alternating Direction Method of Multipliers (ADMM)[33] to the coop-Lasso problem, splitting the variables such that one set is specific to the loss and the other is specific to the penalty, leading to simple solutions for each separate sub-problem. To solve the sub-problem corresponding to the penalty, we use an explicit form of the proximal operator of the coop-Lasso[32]. To speed-up convergence we compute ADMM step-length and over-relaxation parameters in an adaptive fashion[34].

Only the coefficients $\mathbf{B}_{i,\mathrm{CL}}$ are optimized in equation (4). The variance parameters $\sigma_{i,j}^2$ for $j \in C_i$ are treated as known and weights are assumed to be given.

We use a plug-in estimate for the variance parameters, which, when $n_1 > p_r$, is computed using an ordinary least squares (OLS) estimate $\mathbf{B}_{i,\mathrm{OLS}}$ of the regression coefficients of $\mathbf{Z}_{t,1}^{(:, C_i)}$ on $\mathbf{Z}_{r,1}$. Variances are

then estimated in an unbiased way as

$$\sigma_{i,j}^2 = \frac{1}{n_1 - p_r} \left\| \mathbf{Z}_{t,1}^{(:,j)} - \mathbf{Z}_{r,1}\mathbf{B}_{i,\mathrm{OLS}}^{(:,j)} \right\|_2^2 \tag{5}$$

If $n_1 \leq p_r$, a ridge regression estimate is used instead:

$$\sigma_{i,j}^2 = \frac{1}{n_1 - \mathrm{df}_{\mathrm{ridge}}} \left\| \mathbf{Z}_{t,1}^{(:,j)} - \mathbf{Z}_{r,1}\mathbf{B}_{i,\mathrm{ridge}}^{(:,j)} \right\|_2^2, \tag{6}$$

where the ridge penalty is chosen as as the minimal eigenvalue of $\mathbf{Z}_{r,1}^\top \mathbf{Z}_{r,1}$ plus a small fudge factor of $10^{-4}$. The effective degrees of freedom, $\mathrm{df}_{\mathrm{ridge}}$, are used when calculating the variance.

To debias the estimated coefficients in equation (4), weights are selected in a fashion similar to the adaptive Lasso[78] before estimation of $\mathbf{B}_i$. Setting the weights to

$$\mathbf{w}_i^{(k)} = \sqrt{|C_i| / \| \mathbf{B}_{i,\mathrm{OLS}}^{(k,:)} \|_2} \tag{7}$$

improves the selection of regulators.

**Determining cluster membership.** In Step 2, cluster membership is re-allocated, based on the updated cluster structure determined in Step 1. To do so requires the estimation of non-negative coefficients $\mathbf{B}_i$ and residual variances $\sigma_{i,j}^2$ for all clusters and target genes. Previously, these were only estimated for the target genes contained in each cluster. In addition, re-estimation of the coefficients without penalization for the selected regulators and their signs removes the bias the coop lasso introduced and enforces the chosen signs. Finally, the updated cluster membership matrix $\mathbf{\Pi}$ is computed.

Given $R_i$ and $\mathbf{s}_i$, we used sign-constrained linear regression using non-negative least squares (NNLS)[35] to determine the coefficients $\mathbf{B}_i$ for each cluster. To do so, the following optimization problem was solved

$$\mathbf{B}_i = \arg\min_{\mathbf{B}} \left[ \frac{1}{2} \left\| \mathbf{Z}_{t,1} - \mathbf{Z}_{r,1}^{(:, R_i)} \mathrm{diag}(\mathbf{s}_i)\mathbf{B} \right\|_F^2 \right] \text{ such that }, \mathbf{B} \geq \mathbf{0}, \tag{8}$$

where the inequality is considered element-wise. To compute the NNLS coefficients efficiently, we modified an existing algorithm[79] to be able to perform computations on a matrix of responses instead of on a single vector. To avoid unnecessary computation, responses are excluded from the computations once they reach the desired convergence criterion.

The variance parameters $\sigma_{i,j}^2$ for each $i = 1, ..., K$ and all $j = 1, ..., p_t$ are re-estimated as

$$\sigma_{i,j}^2 = \frac{1}{n_1 - |R_i|} \left\| \mathbf{Z}_{t,1}^{(:,j)} - \mathbf{Z}_{r,1}\mathrm{diag}(\mathbf{s}_i)\mathbf{B}_i^{(:,j)} \right\|_2^2 \tag{9}$$

In the case that $n_1 > R_i|$ for any cluster, a warning is issued since this indicates the penalty parameter is set to low and NNLS estimates are unstable.

In the following, computations were performed on the second data split $\mathbf{Z}_{t,2}$ and $\mathbf{Z}_{r,2}$ to avoid bias towards the most complex regulatory programs.

Rag bag clustering is used to identify target genes that do not fit well in any cluster. To do so, the predictive $R^2$-value, denoted as $R_{i,j}^2$, is computed for each target gene and cluster from the residuals of predicting $\mathbf{Z}_{t,2}$ from $\mathbf{Z}_{r,2}\mathrm{diag}(\mathbf{s}_i)\mathbf{B}_i$. The best predictive $R_{i,j}^2$ across clusters for each target gene is recorded. If this best value is below a user-specified threshold $\tau$, then the gene is considered noise and badly predicted within all clusters. It is then placed in a noise cluster/rag bag. Only the remaining target genes are considered in the following steps.

To include prior knowledge of target genes that have a biological relationship, the algorithm allows the user to supply an indicator

matrix $\mathbf{J}$ of size $q \times q$ such that $\mathbf{J}^{(i,j)} = 1$ if genes $i$ and $j$ have a biological relationship, and zero otherwise. It is assumed that $\mathbf{J}^{(i,i)} = 0$ to simplify computations below. By providing gene symbols for the target genes in $\mathbf{Z}_t$ and the genes in $\mathbf{J}$, the algorithm determines the genes for which prior information is available. It is not required that the provided sets are equal as long as there is overlap. Assume for sake of notation that prior information is provided for all target genes in $\mathbf{Z}_t$.

For a fixed target gene $j$, $\mathbf{J}^{(j, :)}$ contains 1's for those target genes which have a biological relationship with gene $j$. Given the current cluster membership matrix $\mathbf{\Pi}$, compute fractions $f_{i,j} = \mathbf{J}^{(j,:)} \mathbf{\Pi}^{(i,:)\top} / \sum_g \mathbf{\Pi}^{(i,g)}$ to encode the biological evidence supporting gene $j$ to be in cluster $i$. In case cluster $i$ is empty, set $f_{i,j} = 0$. These fractions are then normalized across clusters as $p_{i,j} = f_{i,j}/\sum_c f_{c,j}$. For numerical reasons, log-probabilities are used below. To avoid taking the log of zero, a small baseline parameter $\alpha = 10^{-6}$ is added to each $f_{i,j}$ before normalization.

Given $R_i$, $\mathbf{s}_i$, $\mathbf{B}_i$, and $\sigma_{i,j}^2$, the likelihood for target gene $j$ across clusters is computed for observation $l$ as

$$\mathbf{L}_j^{(i,l)} = \frac{1}{\sqrt{2\pi\sigma_{i,j}^2}} \exp\left(-\frac{1}{2\sigma_{i,j}^2}\left(\mathbf{Z}_{t,2}^{(l,j)} - \mathbf{Z}_{r,2}^{(l,R_i)}\mathrm{diag}(\mathbf{s}_i)\mathbf{B}_i^{(:,j)}\right)^2\right). \quad (10)$$

These are then normalized by setting $\mathbf{L}_j^{(i,l)} \leftarrow \mathbf{L}_j^{(i,l)} / \sum_c \mathbf{L}_j^{(c,l)}$. To update the cluster membership for target gene $j$ we then compute votes

$$\mathbf{v}_j^{(l)} = \arg\max_i\left[(1-\mu)\log\mathbf{L}_j^{(i,l)} + \mu\log p_{i,j}\right] \quad (11)$$

for each observation $l$ and assign target gene $j$ to the cluster which receives the majority of votes. The parameter $\mu \in [0,1]$ can be used to control the strength of the prior on the allocation process.

All target genes are processed in a random ordering to avoid introducing bias. Prior fractions for each gene are computed in each iteration using the previously updated cluster assignments as well as old cluster assignments for genes that have not been updated yet.

**Determining convergence.** A history of the cluster membership matrices is kept and the algorithm is stopped when the current $\mathbf{\Pi}$ at cycle $c$ computed in Step 2 is equal to the cluster membership matrix in a previous cycle $c_0$. At this point, the algorithm would enter a loop of length $c - c_0$ and is therefore exited. We consider the algorithm as converged if $c - c_0 = 1$. If $c - c_0 > 1$, the algorithm has found an unstable cluster configuration and results for each possible configuration within the loop are returned.

**Limitations.** Due to the use of unbiased estimates of the residual variance of target genes in Eqs. (5) and (9) it is necessary that $n_1 > p_r \geq |R_i|$ for each $i$.

**Regulatory table.** In addition to all estimated quantities, the algorithm returns a $p_r \times K$ regulatory table $\mathbf{T}$. It is computed as follows for $k = 1, \ldots, p_r$ and $i = 1, \ldots, K$

$$\mathbf{T}^{(k,i)} = \frac{1}{|C_i|}\sum_{j \in C_i}\mathbf{B}_{i,\mathrm{CL}}^{(k,j)} \quad \text{if} \quad \mathrm{median}_{j \in C_i}\left\{\mathrm{corr}(\mathbf{Z}_{t,1}^{(:,j)}, \mathbf{Z}_{r,1}^{(:,k)})\right\} > 0.025 \quad (12)$$

or $\mathbf{T}^{(k,i)} = 0$ otherwise. It summarizes the average effect of an active regulator within a module, given that the regulator is sufficiently correlated with the target genes in that module.

**Internal validation measures**
In the following, we will consider three different types of predictive $R^2$-values. First, predictive $R^2$ per module, written $R_i^2$ for cluster $i$, is the predictive $R^2$ for a given module $i$ computed on the second data split. Second, predictive $R^2$ per module excluding regulator $k$, written $R_{i,-k}^2$,

is the predictive $R^2$ for a given module $i$ conditional on regulator $k$ not being associated with the module. Thirdly, predictive $R^2$ per module and target gene, written $R^2i, j$, is the predictive $R^2$ for target gene $j$ computed within the regulatory program of module $i$. The latter is also called cross-cluster predictive $R^2$ for target gene $j$.

**Guidance on selection of the penalty parameter.** To evaluate the clustering results we introduce two performance scores. The aim of our algorithm is to associate modules with their linearly most predictive regulators. As a measure of clustering quality, it is therefore natural to consider the predictive $R^2$ per module, computed on the second data split with the coefficients $\mathbf{B}_i$ for the selected regulators re-estimated on the first data split as in equation (8). We therefore compute

$$R_i^2 = \left\|\mathbf{Z}_{t,2}^{(:,C_i)} - \mathbf{Z}_{r,2}^{(:,R_i)}\mathrm{diag}(\mathbf{s}_i)\mathbf{B}_i\right\|_F^2 \Big/ \sum_{j \in C_i}\left\|\mathbf{Z}_{t,2}^{(:,j)} - \frac{1}{n_2}\sum_{l=1}^{n_2}\mathbf{Z}_{t,2}^{(l,j)}\right\|_2^2. \quad (13)$$

In addition, we compute the importance of each regulator within a module as follows. The regulator and sign sets without regulator $k$, denoted as $R_{i,-k} =: \{g : g \in R_i, g \neq k\}$ and $\mathbf{s}_{i,-k}$, are used to re-estimate the coefficients for module $i$. We then compute $R_{i,-k}^2$ as the predictive $R^2$ value on this reduced regulatory program and compute the importance of regulator $k$ in module $i$ as $I_{i,k} = (R_i^2 - R_{i,-k}^2)/R_i^2 = 1 - R_{i,-k}^2/R_i^2$. This is the ratio of the semi-partial correlation of the expression profile of regulator $k$ with the expression profiles in module $i$ to the overall $R^2$ of module $i$ with all regulators. A regulator that is more influential in predicting the target genes in module $i$ will have larger importance. Importance values are clipped to the interval $[0, 1]$.

A decrease in $R^2$ per module with an increase in the penalty parameter $\lambda$ is expected, due to selection of less regulators and therefore less degrees of freedom in the linear models associated with each module. Importance is the ratio of the squared marginal correlation of a regulator with the target genes in a module to the overall $R^2$ of that module. This implies that an increase of importance values is expected to concur with a decrease in number of selected regulators. Therefore, importance is expected to increase with an increase in the penalty parameter $\lambda$. Selection of the penalty parameter can therefore be guided by balancing these two scores. Predictive $R^2$ should be high while importance should neither be too low nor too high, since the latter is typically indicative of too few selected regulators.

**Guidance on selection of the number of modules.** Selection of the number of modules can be aided by cross-cluster predictive $R^2$. For each final module's selected regulators and signs, coefficients are re-estimated for all target genes. These are then used to compute the predictive $R^2$ for each target gene in all available modules. Denote these values as $R_{i,j}^2$ for target gene $j$ when computed with regulators from module $i$. If there are $K$ modules, this will result in a $K \times p_t$ matrix. We then define the following score for each target gene $j$ which has been clustered in module $i$

$$a_j = R_{i,j}^2, \quad b_j = \max(0, \max_{c \neq i} R_{c,j}^2), \quad S_j = \frac{a_j - b_j}{\max(a_j, b_j)}.$$

Due to the similarity with the well-known silhouette value[80], we call $S_j$ the *silhouette score for target gene $j$*. The average silhouette score across all target genes gives a rough measure for clustering quality and can be used as a straight-forward tool to compare different module counts. A larger average silhouette score indicates that target genes on average are better located within modules. To determine the optimal number of modules, average silhouette score should be considered jointly with predictive $R^2$ per module. A good clustering achieves high values in both scores.

## External validation measures

To evaluate the performance of simulations, we compared the groundtruth to the estimated clustering in multiple ways.

**Adjusted Rand index and cluster homogeneity.** To determine overall clustering performance, we use the adjusted Rand index[81], a well-known similarity measure between two clusterings.

Since the adjusted Rand index can be misleading if a cluster is split into subclusters or if one cluster is much larger than the rest (as the noise/rag-bag cluster can be if the regulatory signal is weak), we also compute the average cluster homogeneity. The cluster homogeneity is defined as the maximum proportion of genes in a ground truth cluster that are allocated to the same estimated cluster averaged across clusters (akin to an average per-class accuracy in supervised learning).

**Regulator selection.** To answer whether scregclust selects the right regulators for each module we cannot compare regulators associated with each module directly, since estimated modules might not match clearly with the groundtruth. We therefore check correct selection of regulators for each target gene, as these are easily comparable with the groundtruth. For easier comparison we compute two measures:

1. True positive rate, as the proportion of correctly selected regulators,
2. False positive rate, as the proportion of incorrectly selected regulators.

Both of these measures are between 0 and 1 with larger values being better for the true positive rate and smaller values being better for the false positive rate.

## Implementation

The package was implemented in R (R Foundation for Statistical Computing, https://www.R-project.org/, v4.4.1). Computationally expensive parts were written in C++ using the packages Rcpp (https://cran.r-project.org/package=Rcpp, v1.0.13) and RcppEigen (https://cran.r-project.org/package=RcppEigen, v0.3.4.0.2). The packages igraph (https://igraph.org/, v2.0.3) and ggplot2 (https://ggplot2.tidyverse.org/, v3.5.1) were used for visualization. The version of *scregclust* used in this manuscript was v0.1.9.

## Knockdown experiments

**Cell line availability.** All cell lines used for the below described experiments are available from the human glioblastoma cell culture (HGCC) resource https://www.hgcc.se.

**Target gene-editing by RNP complex delivery.** As regulators of state 5, we chose DDR1 and SOX6. To disrupt gene function we delivered U3065MG cells between passage 16–20 with SpCas9 2NLS and target specific multiguide (Synthego Gene Knockout Kit V2). The RNP complex was delivered using 4D-NucleporatorTM X-unit and an optimized protocol with SF Cell line nucleofection kit S (Lonza, V4XC-2032). RNP complex formation was carried out as per the ′ protocol. Briefly, RNP complex formation was carried out by mixing Cas9-2NLS and sgRNA in a molar ratio of 1:9 at room temperature, followed by adding the estimated amount of U3065MG cells dissolved in nucleofection solution and supplement 1. Experimental control used included non-targeting scrambled, Cas9 only, and mock (nucleofection of cells+RNP complex with no electric pulse). Nucleofection for all experiments was done using optimized settings in SF solution and program CA-137, and no major cell death was observed using these conditions. Details of the multiguides used can be found in Table 3.

**Genotyping and Sanger sequencing.** Seven days post-nucleoporation, edited cells and non-targeting scrambled control group were harvested. To evaluate gene editing, genomic DNA was isolated from

### Table 3 | sgRNA sequence information

| Target | Species | sgRNA sequences (multi-guide) |
|---|---|---|
| DDR1 | Human | UGGAGAGCAGUGACGGGGAU |
| | | CUGCAAGUACUCCUCCUCCU |
| | | GGCCCCCGGCAUGCCGUCCC |
| SOX6 | Human | UAUGGGGUGCAGAGGCAGAU |
| | | UUCCCUUGAGGUUAAAUCCU |
| | | AAAUGGAGAGGUGGCUUGCU |
| Non-targeting scrambled control | | GCACUACCAGAGCUAACUCA |

all samples, and PCR amplification of the edited region was performed, followed by Sanger sequencing of the amplicon. Genotyping PCR was performed using Phusion hot start II high-fidelity PCR master mix (Thermo Scientific, F-565L), on an Applied Biosystems miniAmp thermocycler. PCR cycling conditions are: Initial denaturation 980C (5 min, 1 cycle); amplification (30 cycles)-denaturation 980C (1 min), annealing 620C (30 s), extension 720C (30 s); final extension (1cycle) 720C (10 min). PCR amplicons from non-targeting and target-specific samples were purified using the Macherey-Nagel PCR clean-up kit (NucleoSpinTM PCR Clean-up, Cat 740609.250). PCR amplicons were sequenced using a single-end read at Eurofins Genomics (TubeSeq Supreme service, Germany). Analysis of the Sanger sequencing traces was performed using the freely available web tool "Inference of CRISPR edits" ("ICE", https://ice.synthego.com/) by entering the trace files (.ab1 files) from the edited samples and the non-targeting scrambled control. Details of the primers used can be found in Table 4.

**Temozolomide treatment.** Cells from non-targeting scrambled control and target-specific edited samples were seeded to 384 well plates coated (Greiner Blackwell optically clear bottom) with laminin, 7 days post nucleoporation (the editing from different batches of the experiment confirmed efficient knockdown on day 7). Twenty-four hours after seeding, cells were treated with temozolomide over a dose range from 750 μM to 1.25 μM. Treated cells were placed back into the incubator and an Alamar blue reading was taken 96 h after treatment using a CLARIOstar plate reader (BMG Labtech). To know the effect of TMZ on viability fluorescence reading was measured, background subtracted values were normalized to DMSO control, and dose-response curves were plotted using GraphPad Prism.

## Drug combination treatments

**Cell culture.** Primary GBM cell lines U3013MG (female, age 78), U3017MG (male, age 68), U3028MG (female, age 72), U3065MG (male, age 77), U3071MG (male, age 65), and U3180MG (male, age 77) were obtained from the Human Glioma Cell Culture (HGCC) Biobank at Uppsala University[49]. The cells were cultured in a mixture of Neurabasal (Gibco, #21103-049) and DMEM/F12 (Gibco, #31331-028) growth medium in 1:1 ratio, 1x B27, without Vitamin A (Gibco, #12587001), 1x N2 (Gibco, #17502001) and 1% Pen/Strep (Sigma-Aldrich, #P0781). The cells were grown as an adherent culture on laminin-coated flasks (Sigma Aldrich, #L2020-1MG) and were detached for splitting and seeding by TypLE without phenol red (Gibco, #12604039). The growth medium was supplemented with 10 ng/mL recombinant human FGF-basic (Peprotech, #100-18B) and 10 ng/mL recombinant human EGF (Peprotech, #AF-100-15) each passage. The cells were regularly tested for Mycoplasma by either MycoAlert assay (Lonza, #LT07-418) or MycoStrip (InvivoGen, #rep-mys-50).

**Cell viability assay.** U3013MG, U3017MG, U3028MG, U3065MG, U3071MG, and U3180MG cells were seeded on laminin-coated 384-well plates (Thermo Scientific, #142761 or Agilent, #204628-100) with Multidrop 384 dispenser (Thermo Fisher) at a density of 2000 cells/

**Table 4 | Primer sequences used**

| Target:DDR1 | |
|---|---|
| Primer pair for amplicon | Forward 5'-GGACCTCTACTTCCCCTCCA-3' |
| | Reverse 5'- GTACAGCTGGAAGTGAGGAGA-3' |
| Sanger sequencing primer | 5'-GTACAGCTGGAAGTGAGGAGA-3' |
| Target: *SOX6* | |
| Primer pair for amplicon | Forward 5'-GGTAGTTGTTTCATGATCTACACAA-3' |
| | Reverse 5'-GGGGTTTAGAGTGGACCAC-3' |
| Sanger sequencing primer | 5'-CACAAAAATAACTCTAAGGTCATCTATTTC-3' |

well and volume of 0.04 mL/well. The plates were centrifuged for 1 min at 200 x g and incubated for 24 h at 37°C in a humidified atmosphere with 5% $CO_2$. Afterwards, the cells were treated with temozolomide (Selleckchem, #S1237), dasatinib (Selleckchem, #S1021 or MCE, #HY-10181) dissolved in DMSO to a 10 mM stock solution. The drugs were dispensed with D300e digital dispenser (Tecan) without additional dilution of the stock solution. Afterwards, the plates were centrifuged briefly at 200 x g, and were incubated for 72 h at 37°C in a humidified atmosphere with 5% $CO_2$. Cell viability was measured by the Alamar Blue assay (Invitrogen, #DAL1100), which was added 16 h before the readout or by cell confluence (IncuCyte S3). Afterwards, the growth medium was aspirated with Viaflo96/384 (Integra) multichannel pipette, and fresh growth medium was added with Multidrop 384 dispenser. The cells were treated for an additional 72 h or 96 h with a combination of temozolomide and dasatinib in a 7 × 7 dose-response matrix. 16 h prior the endpoint of the reaction, Alamar blue was added into each well of the plates. The reduction of resazurin to resorufin was used as a proxy for the determination of the cell viability. Additionally, the cell confluence (IncuCyte S3) was used as a proxy of cell viability. The raw measurements from both readouts of the treatments were normalized to the DMSO control. The BLISS coefficient and the sensitivity of the combinations were calculated with SynergyFinder[82] in R.

### Reporting summary
Further information on research design is available in the Nature Portfolio Reporting Summary linked to this article.

## Data availability
No new datasets were generated for this paper. The publicly available data used in this study are available in the following repositories: The Curated Cancer Cell Atlas https://www.weizmann.ac.il/sites/3CA/(Bi et al., 2021, Canon et al., 2020, Chen et al., 2020, Couturier et al., 2020, Darmanis et al., 2017, Hovestadt et al., 2019, Hwang et al., 2022, Ji et al., 2020, Kürten et al., 2021, Ma et al., 2019, Neftel et al., 2019, Pelka et al., 2021, Qian et al., 2020, Rendeiro et al., 2020, Riether et al., 2020, Wang et al., 2019, Wu et al., 2020, Zhou et al., 2021), The UCSC Cell Browser https://cells-test.gi.ucsc.edu/?ds=early-brain(Eze et al., 2021), GEO under accession number GSE173278(Leblanc et al., 2022), GSE193884(Leblanc et al. 2022), GSE129730(Ocasio et al., 2019), GSE122871(Weng et al., 2019), GSE156633(Luo et al., 2021), The Neuroblastoma Cell Atlas https://www.neuroblastomacellatlas.org/(Kildisiute et al., 2021), ArrayExpress under accession number E-MTAB-9296(Larsson et al., 2021), The Sequence Read Archive under accession number PRJNA637987(Manno et al., 2021). Additional details about the publicly available datasets can be found in Table 1. Figure data can be found at https://doi.org/10.6084/m9.figshare.25909156. Source data are provided with this paper.

## Code availability
The R-package can be found on GitHub at https://github.com/scmethods/scregclust. The scripts used to produce the figures in the main text and supplementary are publicly available[83].

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

## Acknowledgements
We thank the Swedish Cancer Society (20 0839 PjF S.N.), Swedish Childhood Cancer Fund (2023-0089 S.N., R.J.), Swedish Research Council (2019-03686 R.J. and 2021-03224 S.N.), Swedish Strategic Research Foundation (BD15-0088 R.J., S.N.) and Knut and Alice Wallenberg Foundation (2022.0057 R.J., S.N.) for financial support. Some of the computations were enabled by resources provided by the National Academic Infrastructure for Supercomputing in Sweden (NAISS) at the National Supercomputer Centre (NSC), Linköping University partially funded by the Swedish Research Council through grant agreement no. 2022-06725.

## Author contributions
S.N. conceived the study. I.L. and F.H. developed the algorithm. F.H. wrote the R-package. I.L. and F.H. performed the computational analyses, with support from R.J. and S.N.. G.P. and A.K. planned, and conducted the combination treatment experiments. S.K. planned and conducted the knockdown experiments. I.L. and F.H. wrote the first version of the paper and all authors assisted in editing.

## Funding

## Competing interests
The authors declare no competing interests.
