## [Peer Review File · Nature Communications]

REVIEWER COMMENTS

Reviewer #1 (Remarks to the Author): Expert in brain cancers, neuro-oncology, therapy, and single-cell genomics; co-reviewed with Reviewer #2

Larsson et al. have developed a novel method to reconstruct the regulatory programs using single-cell RNA-sequencing data called single-cell Regulatory-driven Clustering (scRegClust). This method differs from previously published tools such as SCENIC+ which uses single-cell multi-omics data, potentially increasing general usability. It is also claimed to run three times faster than SCENIC+. The key idea is a novel clustering approach that splits the genes into regulators and targets, then clusters the target genes but borrows information from the regulatory genes to improve clustering accuracy.

Initially, they demonstrate the accuracy of their method by using the results obtained when the method is applied to PBMC data and comparing the results to those obtained using SCENIC+. They also show that the regulators identified are enriched in immune cells (Fig 1E). It will be instructive to see a similar comparison with the regulators identified by SCENIC+ in the same dataset.

As a proof of concept, the authors use scRegClust to analyze scRNA-seq data from brain tumors and developing tissues to identify transcription factors and kinases that regulate distinct cell states. They utilize multiple datasets to uncover the regulation of meta-modules, including ones regulated by the transcription factors SPI1 and IRF8.

1. Beyond the fact that SCENIC+ uses scMultiome data while scRegClust uses only scRNAseq data, have the authors identified other reasons behind the partial disagreements between SCENIC+ and scRegClust? Aside from adjusting penalization, is there a way to change cutoff or p values to see if TFs identified by SCENIC+ and not scRegClust could be identified?
2. Overlap with SCENIC is only about 50%, which is not “comparable” as they claim.
3. It would be useful to also annotate the TFs from SCENIC+ that do not overlap with scRegClust.
4. Could they leverage available epigenome datasets for brain tumors to explain the underlying reasons for the difference between SCENIC+ and scRegClust results?
5. Which methods were used to perform the enrichment (bottom half of heatmap) in Figures 3B and 4B?
6. TBX1/Tbet1 is expressed in DCs, NK and NKT, B cells, and T cells, not just NK cells. Why is it strongly associated with only NK cells?
7. What’s the relationship between “modules” and “clusters” in Fig 3? It is not clearly explained.
8. A well-established predictor of TMZ response is MGM methylation status. Has this been tested and cross-correlated with their finding?
9. Also, PR cells (which are enriched in OPC/NPC-like cells in sc GBM classification in the Neftel study) are more sensitive to TMZ treatment. Why is blocking the transition to state 5 which seems to correspond to OPC-like cells make them more sensitive to TMZ? Could it be that the original study was performed on a cell line, U3065MG, and state5 reflects a more undifferentiated/stem-like cell state in this cell line?
10. Their observation is incongruent with many previously published works. They should test their predictions in primary GBM cells, not a single cell line. Minimally, they should use GBM sc datasets to make their predictions.
11. Did they perform the analysis shown in Figure 4 by first identifying different cell types from each data

set (cancer, normal neural, immune, endothelium, et) and extracting the same cell types first before performing the analysis? Different transcription factors can have different functions depending on the cell type. For example, the microglia signature in module 5 may come from microglia.

12. The correlation between MES glioma cell state and immune signatures is well established in previous studies.

Comments regarding the R Package:

The R package for scRegClust is fairly easy to use and has the potential to provide good insights into important modulators of cell states/subtypes from scRNAseq data.

1. The authors should make it clear that the modulator list is only available for human datasets and that users will need to convert the row names of their expression matrices to humans prior to initiating the analyses.
2. It would be useful if the authors allow the “manual” addition of other regulators in addition to TFs and Kinases.
3. While testing the package, it performed well when using only the 2000 most variable genes, however, it crashed when attempting to test the entire matrix. Similarly, the package won't run if the number of cells is lower than the number of regulators.
4. It appears that the package performs well with TFs and kinases that are highly expressed in a large number of cells. How about important regulators that are expressed by a smaller number of cells or at a lower level?

Minor comments

1. The figures are referenced in the text out of order. For example, Figure 2A and sup Figure S6
2. Some of the legends do not adequately describe the figures.
3. Figure 3M module numbers should be labeled.
4. Figure 4 legend colors are not distinguishable, another color scheme with colors further apart on the spectrum would be more advisable. Also, the row names of the top heatmap need to be resized to prevent overlap. Where is the legend for color-coded heading (different modules???)?

Reviewer #2 (Remarks to the Author): Expert in brain cancers, neuro-oncology, therapy, and single-cell genomics; co-reviewed with Reviewer #1

Reviewer #3 (Remarks to the Author): Expert in bioinformatics, statistics, single-cell regulatory networks

and omics

The manuscript presents scRegClust, a computational approach and software for clustering genes by their co-regulated expression patterns and identifying regulators of these gene clusters in the context of brain cancer. The method proposes that gene expression relies linearly on a limited, cluster-specific subset of regulators. The authors employed a combination of heuristics to solve this optimization problem, incorporating both existing and novel software tools. They validated scRegClust's performance and utility using a range of simulated and real datasets, including up to 1M cells.

Nonetheless, I have several reservations. The manuscript offers scant discussion or comparison with current methods. Its broader significance is somewhat unclear, as the approach is solely demonstrated on brain cancer. The model incorporates multiple hyperparameters, and an inappropriate value choice could significantly impair the method's performance.

Major concerns:

1. Gene clustering and regulator prediction are well-established in scRNA-seq research. Although this study's innovation lies in simultaneously addressing these issues, users would still be better off choosing a separate method for each question if they individually perform better than scRegClust. The absence of comparisons with existing techniques for these two areas is notably troubling. While the paper briefly mentions various gene regulatory network reconstruction methods, it seldom discusses individual gene regulation links. The comparison with SCENIC+ appears barely relevant. To properly position scRegClust, the authors should conduct comprehensive comparisons with established methods in gene clustering and regulator prediction separately, particularly focusing on statistical performance.
2. Linear predictive model results are influenced by the covariance matrix, which encompasses various factors including actual gene regulation, co-regulation, co-variation across cell types, and technical variation. In scRNA-seq, true gene regulation is often a minor element. If the model only considers gene regulation and neglects other factors, finding true positives can become challenging. A potential validation approach is to select an equal number of non-TF, non-kinase genes as regulators. These could be random, highly expressed, or highly variable genes. Would their gene clustering performance, measured by R2, ARI, and silhouette score, be comparable to that achieved using TF and kinase genes?
3. The study's significance is currently limited, as both applications focus on brain cancer. Can the authors demonstrate its utility in a different scenario?
4. ScRegClust involves several user-defined hyperparameters, such as lambda, the minimum size for non-empty clusters, and the initial value for K. Unreported and inconsistent hyperparameter values might lead to unstable performance and irreproducibility, as evidenced in Fig 2AB. Could the authors list all hyperparameters and their values for each analysis in this study? Furthermore, can they propose a strategy for selecting hyperparameters in typical use cases lacking ground truth, even if it's as simple as default values? Does this strategy consistently yield satisfactory results against ground truth in a wide range of simulations, varying in data dimensions, module size, regulator count, and coefficient distribution? The supplementary material's simulations address some of these points but with limited

variation in simulation parameters and focused only on lambda.

5. The current simulation methodology lacks detail. Could the authors mathematically describe their simulation methodology? Does it include a discrete sampling process and produce scRNA-seq read counts?

6. Fig S1B, 1DE: Employing a hypergeometric test could enhance result interpretation. Please report sample sizes, odds ratios, and P-values for each test. Also, clarify the definition of the background population in the Methods section.

7. How are results integrated in Fig 4A?

8. The software can be installed. The tutorial can be partially run with errors quoted below. The outputs from the figures were not entirely error-free, and only a variant of Fig 2A was reproducible, albeit with discrepancies from the paper's version. The tutorial also resulted in over 70% of genes falling into the 'rag bag' cluster. The tutorial should represent a realistic use case and performance, and the authors need to clearly communicate anticipated outputs.

Errors encountered:

```
> plot_regulator_network(fit)
```

```
Error in base::colSums(x, na.rm = na.rm, dims = dims, ...) :
```

```
'x' must be an array of at least two dimensions
```

```
> for (i in 1:5){
```

```
regulator_importance[,i] <- rowMeans(fit$results[[i]]$output[[1]]$importance, na.rm = TRUE)
```

```
}
```

```
Error in regulator_importance[, i] <- rowMeans(fit$results[[i]]$output[[1]]$importance, :
```

```
number of items to replace is not a multiple of replacement length
```

Minor concerns:

The use of $\sigma_{i,j}$ is confusing, as it seems independent of i .

For regulatory i not selected for cluster j , is it regarded as a target in cluster j ? This should be communicated and justified clearly in Methods.

Fig 1C: What criteria were used by the authors to select specific regulators for visualization?

Phrases like "Improving accuracy" as in "Regulatory-driven clustering improves accuracy" can be misleading. The authors should reword such claims to avoid confusion with improvements over existing methods. Additionally, any claims of improvement, even over initial conditions, should be substantiated through comprehensive comparisons.

The manuscript alternates between "penalty parameter" and "penalization parameter" – consistency in

terminology would be beneficial.

The paper made several mention of regulatory network, but scRegClust primarily reconstructs bipartite graphs, which lack several key features of typical regulatory networks, like perturbation propagation through chain reactions. The authors should clearly justify why scRegClust reconstructs regulatory networks, or consider other terms.

Point-by-point response, reviewers 1 and 2

Larsson et al. have developed a novel method to reconstruct the regulatory programs using single-cell RNA-sequencing data called single-cell Regulatory-driven Clustering (scRegClust). This method differs from previously published tools such as SCENIC+ which uses single-cell multi-omics data, potentially increasing general usability. It is also claimed to run three times faster than SCENIC+. The key idea is a novel clustering approach that splits the genes into regulators and targets, then clusters the target genes but borrows information from the regulatory genes to improve clustering accuracy.

Initially, they demonstrate the accuracy of their method by using the results obtained when the method is applied to PBMC data and comparing the results to those obtained using SCENIC+. They also show that the regulators identified are enriched in immune cells (Fig 1E). It will be instructive to see a similar comparison with the regulators identified by SCENIC+ in the same dataset.

As a proof of concept, the authors use scRegClust to analyze scRNA-seq data from brain tumors and developing tissues to identify transcription factors and kinases that regulate distinct cell states. They utilize multiple datasets to uncover the regulation of meta-modules, including ones regulated by the transcription factors SPI1 and IRF8.

1. Beyond the fact that SCENIC+ uses scMultiome data while scRegClust uses only scRNAseq data, have the authors identified other reasons behind the partial disagreements between SCENIC+ and scRegClust? Aside from adjusting penalization, is there a way to change cutoff or p values to see if TFs identified by SCENIC+ and not scRegClust could be identified?

Response. Thank you for the careful assessment of our manuscript. Regarding question 1, we have now carefully analyzed the sources of discrepancies between the two methods and systematically analyzed how the thresholds and penalization affect the overlap. We have also extended our discussion of how scRegClust relates to and complements other clustering and regulator identification methods (discussed in the response to Reviewer 3).

First, we noted that a crucial difference between the two methods lies in the standard preprocessing software used. SCENIC+ is a Python-based software, that relies on Scanpy, whereas scRegClust is R-based and relies on Seurat for preprocessing. As was noted in a recent bioRxiv preprint [1], the Scanpy and Seurat pipelines have surprisingly limited overlap in the selection of highly variable features (Jaccard index 0.22) when run on the same data set using default settings. The reason for this appears to be that the two programs use different default algorithms for variable gene selection. Checking whether this applies to our analysis, we noted that 8 of the 23 regulators found by Scanpy/SCENIC+ code but not by scRegClust were indeed absent from the input matrix to scRegClust, generated using Seurat at an inclusive threshold. Thus, a third of the missing SCENIC+ regulators in the submission version of the paper was attributed to a difference in R vs Python standard pipelines and not the algorithm per se. Accordingly, in all extended analysis below and in the revised paper, we made sure that

all regulators identified by SCENIC+ were part of the input matrix to scregclust, substantially improving the overlap.

Second, as the reviewer suggests, the overlap with SCENIC+ can be increased even further by adjusting the penalty parameter. To demonstrate this, we ran scregclust for a wide range of penalty parameters (from 0.0001 to 0.5) and quantified the overlap with the regulators identified by SCENIC+ (please see figure below):

It's clear from this figure that when the penalty parameter is low enough, scregclust identifies all regulators that SCENIC+ does. However, the total number of regulators identified by scregclust at these low penalizations is higher, as evidenced in the figure below. Once the penalty parameter gets harder, there is a drop in the number of identified regulators (very clear at 0.05 in the figure below). Ultimately, the statistical significance of the overlap of the regulators between the two methods is the crucial aspect, see next question.

To summarize, we do identify all regulators that SCENIC+ does if the penalization is low enough (and all regulators are included in the input matrix), but at the expense of the regulatory model being too inclusive at these penalties. In the relevant range (0.1-0.2, as evidenced by the diagnostic plots below) we have an overlap with SCENIC+ of 60-40 %. **We**

have updated **Figure 1 C-E** with the latest run, at a penalization = 0.1.

2. Overlap with SCENIC is only about 50%, which is not “comparable” as they claim.

Response. Thank you for the comment. We have statistically assessed the overlap between the two methods using Fisher’s exact test and find that the overlap represents an odds ratio of 30 with a p-value of magnitude <10E-16. The text has now been rephrased to reflect these observational facts (**page 6**). We opted for the phrase ‘strong association’, which we think is a fair reflection of OR=30.3711. As discussed above, the choice of penalty parameter can be tuned, allowing the user to choose between sparse and complex solutions.

3. It would be useful to also annotate the TFs from SCENIC+ that do not overlap with scRegClust.

Response. Thank you for this suggestion. We have extended the annotations to include regulators only found by SCENIC+ (using the Python-based Scanpy preprocessing). The results can be found in an updated version of **Supplementary Table 1**, and we have adjusted the text in the manuscript accordingly, see **page 6**. In short, this analysis showed that a larger fraction of the scregclust-specific regulators were immune-cell enriched/enhanced (81 %) compared to the SCENIC+ -specific regulators (43 %).

4. Could they leverage available epigenome datasets for brain tumors to explain the underlying reasons for the difference between SCENIC+ and scRegClust results?

Response. Thank you for suggesting this. We would like to clarify that the SCENIC+ comparison at the beginning of the paper was focused on the commonly used PBMC data set

generated by 10X Genomics, i.e. not a neural cancer but blood cells. As discussed above, key factors for the difference between results are the data preprocessing, hyperparameter tuning, and presence/absence of ATAC data, which we can expect to apply to several contexts, including neural cancers. In our extended analysis (**Figure 4 and newly added Figure 5**), we have opted to process an extensive collection of data sets from both neural and non-neural cancers, leveraging the speed of our method to map possible regulators across tumor types; see below. This serves to illustrate the power of the proposed modeling method.

5. Which methods were used to perform the enrichment (bottom half of heatmap) in Figures 3B and 4B?

Response. Thank you for pointing this out. We used Jaccard Index to quantify the similarity between our module gene sets and gene signatures of interest, as has been done in e.g. Chanoch□Myers et al. 2022, <https://doi.org/10.1186/s13073-022-01109-8>. This has now been added, see e.g. **Results page 9, Methods page 16 and Figure 3 and 4 legends**.

6. TBX1/Tbet1 is expressed in DCs, NK and NKT, B cells, and T cells, not just NK cells. Why is it strongly associated with only NK cells?

Response. We thank the reviewer for this comment. We respond assuming that the question refers to TBX21/Tbet1 (as opposed to TBX1, a different TF). We have improved the data presentation to highlight that TBX21/Tbet1 is also detected in other cell types, by including more output information from the scregclust analysis (**c.f. new Figure 1C**). Figure 1C now clearly shows that TBX21 most strongly regulates a module that enriches for NK signature genes, but also CD8 T cells (module 7). In addition, it has a weaker (but non-zero) interaction with a module enriching for T-cell signatures in general (module 1). By studying the SCENIC+ results for this particular regulator (SCENIC-paper, Figure 2C), we can see that the most robust regulation is indeed towards NK-cells, followed by CD8 T-cells and a weak signal in CD4 T-cells. Neither of the methods detected TBX21 as a regulator for DCs for this particular data set. An independent data source, the Human Protein Atlas, notes an enrichment for NK cells. We propose that running the algorithms on another/a larger data set encompassing more DC/B-cells might be required to find these additional associations.

7. What's the relationship between "modules" and "clusters" in Fig 3? It is not clearly explained.

Response. Thank you for pointing this out. We agree that these concepts might cause some confusion. Technically because *scregclust* finds sets of co-regulated gene sets, it can be viewed as an algorithm that (as part of its output) clusters genes. In the single-cell field, however, the word cluster is most commonly used to denote a group of cells with similar gene expression profiles. Therefore, we opted for the term "gene modules" for co-regulated genes found by *scregclust*. The term gene module is often used to denote a set of co-regulated genes and we think it is more intuitive than the many alternatives (regulon, signature, gene battery, and so forth). **We have now carefully revised the text with this distinction in mind.** Figure 3 in the revised paper does not use the term cluster.

8. A well-established predictor of TMZ response is MGM methylation status. Has this been tested and cross-correlated with their finding?

Response. Thank you for your comment. We have now performed these tests and analyses. We agree that substantial literature provides correlative evidence that MGMT methylation can predict TMZ response, and mechanistic evidence shows that methylation of the MGMT promoter reduces levels of MGMT, a DNA repair enzyme, thereby promoting TMZ sensitivity [2]. One study showed that variation in MGMT promoter methylation explained up to 46% (correlation of 0.68 squared) of the variability in TMZ response in cell lines, although this percentage could be lower depending on growth conditions [3]. This suggests that additional factors may also be important for the TMZ Response.

In our paper, we used patient-derived primary GBM cells from our Uppsala biobank HGCC, called U3065MG. We focused on U3065MG because we have previously combined single-cell analysis, barcoding, and mathematical modeling of this culture to identify a specific transient cell state (termed "state 5" in Larsson et al., *Molecular Systems Biology* 2021) associated with relative resistance to TMZ. To showcase scregclust, we sought to identify kinases and transcription factors that could be targeted to suppress state 5 and increase TMZ sensitivity. Inspection of methylation data from our HGCC biobank shows that this culture has a methylated MGMT promoter (<https://hgcc.se/#data>). Additionally, there is no variation in MGMT expression among U3065MG cells in the scRNA-seq data from Larsson et al [4] - MGMT is not detected in any cell. We conclude that U3065MG cells are consistently MGMT-methylated and that any variation in the TMZ response between cell states is unrelated to MGMT methylation status. This clarification has now been added to the manuscript (**Results, p 10**).

Further exploring this theme in the revised paper, we investigated whether the synergistic combination identified in U3065MG cells is equally potent in additional primary patient-derived cell cultures with different MGMT methylation statuses. Accordingly, we selected an additional five primary cell cultures from the previously mentioned HGCC database, with variable MGMT methylation, and tested our combination treatment regimen of 72-hour pre-treatment with dasatinib followed by 96 hours of combined treatment with dasatinib and TMZ. By relating the mean average bliss score (synergy score) to methylation status, we observed a trend of increased synergy if the cell culture is methylated (Figure S5). However, the difference was not statistically significant, indicating that MGMT methylation likely contributes to the treatment effect in terms of cells being more responsive to TMZ, but it is not the main explanatory factor for the identified synergistic combination.

These updates are reflected in Figure S5 and described in the main manuscript, page 10-11.

9. Also, PR cells (which are enriched in OPC/NPC-like cells in sc GBM classification in the Neftel study) are more sensitive to TMZ treatment. Why is blocking the transition to state 5 which seems to correspond to OPC-like cells make them more sensitive to TMZ? Could it be that the original study was performed on a cell line, U3065MG, and state5 reflects a more undifferentiated/stem-like cell state in this cell line?

Response. Thank you for your comment. We are assuming that "PR" refers to the proneural subtype as described by [5], not the PPR subtype as mentioned by [6]. The original study utilized a patient-derived primary cell culture, U3065MG. The rationale for suppressing state 5 is explained in Larsson et al., *Molecular Systems Biology* 2021. According to the STAG model—which was fitted to longitudinal data collected over three weeks and provided the net rate of growth for each state along with transitions between states—it was predicted that the simplest effective intervention in U3065MG cells was to suppress transitions to state 5. As detailed in the paper, the STAG model accounts for state-specific differences in growth rates, and the 'stemness' of each state is assessed by its position within the network. Our findings indicated that state 5, while enriching for the OPC-like signature, does not enrich for the proneural signature. Furthermore, state 5 is less proliferative than several other states and lacks the proliferative markers or stemness markers that other, more undifferentiated/stem-like cells exhibit.

The central point of the U3065MG example in the current paper is highlighting the modulation of drug-sensitive states as a potentially significant application of scregclust. In this context, we demonstrated how scregclust integrates into a workflow for making predictions about modifying state transitions. The actual problem motivation (blocking the transition to state 5 to enhance TMZ treatment) is more thoroughly discussed in the previous paper.

We have revised the data presentation to make clear that the U3065MG example is to highlight a particular application of scregclust, done in **Results page 8 and Discussion page 14**.

10. Their observation is incongruent with many previously published works. They should test their predictions in primary GBM cells, not a single cell line. Minimally, they should use GBM sc datasets to make their predictions.

Response. Thank you for the comment. We would like to clarify that the original submission did involve a patient-derived GBM culture and that we tested a prediction derived from a single cell data set from patient-derived GBM cells. Thank you for the suggestion to extend the experimental tests. As detailed above, while the original prediction applied to 3065MG cells, we have tested the prediction in five more patient-derived cultures, discussed above. In our view, the targeting of a gene signature found by dynamic analysis of longitudinal single-cell data is a new and interesting application. We have carefully worded the revised paper to emphasize that we bring up this example as a use-case for the algorithm that opens for extended work.

11. Did they perform the analysis shown in Figure 4 by first identifying different cell types from each data set (cancer, normal neural, immune, endothelium, et) and extracting the same cell types first before performing the analysis? Different transcription factors can have different functions depending on the cell type. For example, the microglia signature in module 5 may come from microglia.

Response. The analysis was conducted separately on each dataset; subsequently, the modules were merged into one regulatory table and visualized in the same heatmap in **Figure 4B**. For the patient-derived tumor datasets, non-malignant cells were filtered out based on

metadata from each respective study or copy number aberration (CNA) analysis. In the analysis underlying Figure 4, any module derived from a dataset representing a patient tumor (annotated as GBM, MB, or NB) is composed solely of malignant cells. The enrichment of a microglia-related signature for that specific disease module is not due to the module representing microglia cells but instead suggests a previous interaction between the malignant cells and microglia, as discussed in the paper. Conversely, if the module is derived from a normal (non-malignant) developing brain dataset (annotated as Normal), the enrichment of the microglia signature indeed indicates that the module represents microglia cells. It is important to note that meta-module 5 consists of a mix of modules derived from both disease datasets and a normal developing brain (Figure 4B, Box 1) and that the enrichment of microglia signatures occurs in both types of modules. This is very interesting, as it shows that tumor cells can adopt the regulatory programs that are usually active only in immune cells, a phenomenon previously described as myeloid mimicry by [7] (referenced in the manuscript).

Regarding the statement that different transcription factors can have varying functions depending on the cell type, we agree. Nevertheless, we expect that scregclust can capture these nuances even if the analysis is not run separately on each cell type (cancer, normal neural, immune, endothelium). This would be evident in the output as one transcription factor regulating several modules, each with distinct functions and possibly varying modes of interaction (activating or repressing).

The above points have been clarified throughout the manuscript.

12. The correlation between MES glioma cell state and immune signatures is well established in previous studies.

Response. We agree with the reviewer that this topic has been discussed in previous studies. The new point we sought to make is that our model identifies a possible mechanism behind the immune cell-induced mesenchymal shift, showcasing how our method can be used to derive new biological hypotheses. In the initial submission, we did reference previous studies on this topic in the Introduction and now emphasize this also in the **Results section (page 11)**.

Comments regarding the R Package: 1. The authors should make it clear that the modulator list is only available for human datasets and that users will need to convert the row names of their expression matrices to humans prior to initiating the analyses.

Response. We thank the reviewer for bringing this to our attention. It is correct that the package is distributed with lists for human regulators (transcription factors and kinases). That said, the user is entirely free to provide the algorithm with any custom regulator list (e.g. FDA-approved drug targets or similar). If the data is murine, this could for instance be a set of mouse transcription factors (c.f. <https://www.nature.com/articles/ncomms15089>). We have clarified this in the manuscript and user information (**Methods page 16**).

If the user wishes to add regulators manually, this is done by modifying the input vector “is_regulator.” This is an indicator vector, telling which rows in the expression matrix should be

considered regulators, and the user can make any custom assignment they wish. Of note, the scregclust algorithm itself, and the main function scregclust, are indifferent to species or regulator type, it is only the convenience function scregclust_format that comes with pre-compiled lists of human transcription factors and kinases, as we expect this to be the most common scenario.

2. It would be useful if the authors allow the “manual” addition of other regulators in addition to TFs and Kinases.

Response. Please see above. This is already a feature. The user is entirely free to provide the algorithm with any custom regulator list and instructions for this are provided in the package. As a long-term goal, we will expand on the set of regulator panels available in the distributed software, also addressing multi-species and cross-species functionality.

3. While testing the package, it performed well when using only the 2000 most variable genes, however, it crashed when attempting to test the entire matrix. Similarly, the package won't run if the number of cells is lower than the number of regulators.

```
r$> fit <- scregclust(
  z[, 1:100], genesymbols, is_regulator,
  penalization = exp(seq(log(0.05), log(0.2), length.out = 5)),
  n_cl = 10L, n_cycles = 1L, noise_threshold = 0.05, center = FALSE,
  sample_assignment = sample_assignment[1:100]
)
→ Validating input
Error in `scregclust()`:
! Too few cells.
× Each data split needs to contain more cells than there are regulators.
i Number of regulators = 491
i Elements in split 1 = 50
i Elements in split 2 = 50
Run `rlang::last_trace()` to see where the error occurred.
```

Response. Thank you for testing the algorithm. Two issues are mentioned. The first is that the algorithm requires the number of cells (n) to be greater than the number of regulators (p). The second is that the program 'crashes' on the reviewer's computer when running the full matrix. The first issue is not something we usually encounter as for real-life use cases, the number of cells (n) is often substantially larger than that of regulators (p). For instance, a 10X scRNAseq run generates approximately $n = 8,000-10,000$ cells. One possible workaround, e.g. when analyzing smartSEQ or similar data sets, is to make an informed pre-selection of important regulators to reduce their quantity. The requirement that $n > p$ (more cells than listed regulators) is linked to both Steps 1 and 2 of *scregclust*. In Step 1, we use an ordinary least squares (OLS) estimate of the residual variance for each target gene using all regulators as predictors. However, since this quantity is only used to scale the data we have now added an alternative estimator of residual variance based on ridge regression so that the package can be used even with $n < p$. Similarly, in Step 2 (the non-negative least squares (NNLS) estimation), the number of regulators used during estimation depends on the cluster. For low penalization parameters, all p regulators may be involved. Even though NNLS can technically be performed even when $n < p$, a unique solution requires $n > p$. Here, we have added a

warning message to the package, stating that when the number of selected regulators exceeds n , NNLS can perform poorly and the penalization parameter has been set too low.

Regarding the second issue, it appears to be a memory issue on the reviewer's machine. The algorithm has been observed to consume memory (RAM) proportional to the size of the input data. Without the exact error message and further information on the runtime environment, we cannot definitively explain why this occurs for the reviewer. We have added a clarification in the text that the machine used must contain enough memory to handle the data set at hand. The algorithm is efficient in the sense that all calculations in our original and revised paper were conducted on a laptop (MacBook Pro, M1, 16 GB RAM).

We have added a clarification to the manuscript (**Results p 5, Methods p 24**).

4. It appears that the package performs well with TFs and kinases that are highly expressed in a large number of cells. How about important regulators that are expressed by a smaller number of cells or at a lower level?

Response. We thank the reviewer for this perceptive comment. To understand how well our algorithm can detect regulators that are expressed in a smaller number of cells, we chose a few representative scregclust-runs (on real datasets of differing sizes and from different cancer types) and investigated the relation between the percentage of cells the regulator is expressed in vs the effect size for the fitted models' regulators. Three examples can be seen below. If it would be essential for the algorithm that the regulator is expressed in a large number of cells, we would expect to see a clear positive correlation between effect size and percentage of cells, as well as a sharp drop in detected regulators when the percentage of cells goes below a certain point. As can be seen in the figures below, neither of these two scenarios are observed. Rather, the effect size and number of detected regulators seems to be stable going from regulators expressed in a low percentage of cells to regulators expressed in the majority of cells in the dataset.

Minor comments 1. The figures are referenced in the text out of order. For example, Figure 2A and sup Figure S6

Response. Thank you for drawing our attention to this. We have updated our figure numbering to better align with the text.

2. Some of the legends do not adequately describe the figures.

Response. We have carefully revised the figure legends to align with the figure and meet the journal's style.

3. Figure 3M module numbers should be labeled.

Response. Thanks for pointing this out. We assume that the reviewer means Figure 3B. Module numbers have been added.

4. Figure 4 legend colors are not distinguishable, another color scheme with colors further apart on the spectrum would be more advisable. Also, the row names of the top heatmap need to be resized to prevent overlap. Where is the legend for color-coded heading (different modules???)?

Response. We agree and have updated the figure accordingly. We have also clarified in the figure legend that the color-coded middle heading references the meta-modules described in Box 1.

Point-by-point response, reviewer 3

1. Gene clustering and regulator prediction are well-established in scRNA-seq research. Although this study's innovation lies in simultaneously addressing these issues, users would still be better off choosing a separate method for each question if they individually perform better than scRegClust. The absence of comparisons with existing techniques for these two areas is notably troubling. While the paper briefly mentions various gene regulatory network reconstruction methods, it seldom discusses individual gene regulation links. The comparison with SCENIC+ appears barely relevant. To properly position scRegClust, the authors should conduct comprehensive comparisons with established methods in gene clustering and regulator prediction separately, particularly focusing on statistical performance.

Response. Thank you for the careful assessment of our manuscript. We have extended the presentation, motivation of the approach, and technical evaluations. First, the reviewer will likely agree that while the clustering of cells is commonplace in scRNA-seq, less has been said about clustering of genes. As an example of the current emphasis in the field, a recent review by Fabian Theis and colleagues [8] elaborates substantially on cell clustering but does not mention gene clustering. The same review also mentions SCENIC+ as a go-to method for estimating regulators of cell states with combined scATAC and scRNA data. We agree with the reviewer that scRegClust and SCENIC+ are different (crucially, the input data), but they have a similar biological intent, and for that reason we thought it is a good comparison that readers will be interested in.

We agree that methods have been proposed to build gene-gene association networks for single-cell data. However, due to the high noise and sparsity of single-cell transcriptomic data, the effect of technical variation could be more critical for single-cell co-regulatory network inference. An evaluation of coexpression-based network inference using scRNA-seq data from 31 individual studies comprising 163 cell types showed lower retrieval of known functional

links compared to bulk RNA-seq data [9]. This uncertainty in network estimation underscores the importance of models with fewer degrees of freedom. The number of parameters in a gene network with n genes grows by n^2 , whereas a module network has only $(1+k)n$ parameters, where $k \ll n$ is the number of modules (typically around 5-10, with n in the thousands). Despite having 1-2 orders of magnitude fewer parameters, the module network fitted by *scregclust* retains the key links of interest, typically between TFs or kinases and co-expressed genes with similar functions. As demonstrated, this approach extracts meaningful insights from datasets by selecting a sparse, actionable set of regulators per process and aligning processes across multiple datasets and diagnoses, providing a comprehensive overview. Users can flexibly change the set of regulators beyond TFs and kinases.

scregclust is not intended for clustering genes per se, but for identifying correlated gene blocks best explained, in a sign-consistent manner, by a limited set of regulators. It is designed to avoid creating a complete network with an unfeasibly high number of parameters, focusing instead on how key regulators affect major cellular processes.

Despite these important differences in concept and scope, three types of technical comparison were possible:

- First, focusing on the task of robustly clustering genes, we compared *scregclust* to the most commonly used clustering algorithms. Here, *scregclust* performs better than hierarchical clustering and shows more consistent behavior compared to k-means when applied to 10X data from PBMCs and brain tumors (**Supplementary Figure S8**). This is reassuring, but we point out that *scregclust* provides more information than clustering methods do. As explained in the Supplementary material, clustering stability was measured by Adjusted Rand Index; genes were randomly broken into three groups. Each algorithm was run on Part 1 + 3 and then Part 2 + 3. Stability of clustering was evaluated by comparing the clustering of Part 3 between the two runs repeated 50 times.
- Second, focusing on the task of robustly finding regulators linked to clusters, we compared *scregclust* to network construction methods combined with community detection. Specifically, we included partial correlation-based networks, PPCOR, and weighted correlation networks, WGCNA. Ours and WGCNA perform best in terms of stable selection of regulators. However, WGCNA tuning performance appears to vary a lot across the two data sets, and fewer regulators exhibit a stable selection performance. Similarly, the partial correlation network (PPCOR) is very sensitive to tuning parameter settings, and is very unstable in terms of selection performance, especially for the PBMC data (**Supplementary figure S9, S10**). In addition, PPCOR and WGCNA cannot allocate a regulator to more than one module, thus preventing the detection of cell-state specific regulatory programs. **As explained in the Supplementary Material**, the stability metric was computed by finding all edges between regulators and genes in Part 3 above predicted by a method, each method for a range of parameters, and computing the proportion of times across 50 runs a regulator is associated with at least one of the modules. We also compare the stability of regulator identification with GRNboost2. Here, regulator identification is, by con-

struction, conducted separately for each target gene. Thus, when we cluster target genes there is no guarantee that the same regulators are selected for all targets in the same cluster. This is illustrated in **Supplementary figure S11**), where we plot the proportion of target genes in a cluster that share a selected regulator against the number of clusters.

- Third, we assessed regulator sign-consistency within clusters. The main idea behind *scregclust* was to make sure genes in a clusters are influenced by the identified regulators in a consistent way. To assess this, we calculated the signs of the regulatory effects on genes in Part 3. Because one regulator can affect multiple genes in a cluster, we examined the distribution of signs for that regulator within the cluster. We then plotted the proportion of the most common sign (+1 or -1) against the number of connections to that regulator in the cluster. In *scregclust*, this process is integrated into the method, ensuring that sign consistency within clusters is always achieved with a majority sign proportion of 1. Other methods (PPCOR, WGCNA, and kmeans with GRNboost2) don't have a straightforward way of doing this. One approach used in other methods, such as SCENIC+, is to consider the sign of the marginal Spearman correlation between a regulator and a gene. This was the method used for the other three approaches we compared (**Supplementary figure S12, S13**). However, it's clear that these methods don't guarantee sign consistency within clusters, meaning target clusters contain genes that are not regulated by the identified regulators in the same way.

Taken together, *scregclust* has a different scope than clustering and network construction methods. When used for clustering, *scregclust* shows top-level performance. *Scregclust* also outperforms 'naive' combinations of clustering and network construction. These points are now made in Results (**page 8**) and Supplementary Material)

2. Linear predictive model results are influenced by the covariance matrix, which encompasses various factors including actual gene regulation, co-regulation, co-variation across cell types, and technical variation. In scRNA-seq, true gene regulation is often a minor element. If the model only considers gene regulation and neglects other factors, finding true positives can become challenging. A potential validation approach is to select an equal number of non-TF, non-kinase genes as regulators. These could be random, highly expressed, or highly variable genes. Would their gene clustering performance, measured by R2, ARI, and silhouette score, be comparable to that achieved using TF and kinase genes?

Response. Thank you for this comment. First, we note that the biological goal of this paper is to identify TFs and kinases that best explain the plasticity of neural and non-neural cancers. This is an unaddressed problem, where strong methods are needed that process the data using principled algorithms. An important prior, or premise, for our analysis is that capturing TFs and kinases is intrinsically interesting. The TFs are of particular interest for capturing ways in which the cancer co-opts developmental, differentiation, or wound healing mechanisms. The kinases are of particular interest given their proven feasibility as pharmacological targets.

In the paper, we illustrate how scregclust uses this prior to link TFs, kinases and processes to illuminate several neural cancer diagnoses, as well as carcinomas and leukemias. The method can thus be a starting point to pick genes for further experimentation, rather than as a general feature selection tool. These points have now been clarified further. The user is free to define the `is_regulator` vector of the package to deal with any gene set as possible explanatory variables. We are not trying to estimate the optimal set of predictive genes for other genes. This would be a suboptimal approach and mainly correspond to a clustering or regulatory network setup, and thus associated with instabilities. Our focus is here to allow a user to input actionable units that can be used for experimental follow-up and gear the study based on identified regulatory programs. There may, of course, be both a random assignment and optimal assignment of target genes that are better predictors for other target genes than TFs/kinases due to their pairwise correlation, but these might not be good candidates for intervention.

3. The study's significance is currently limited, as both applications focus on brain cancer. Can the authors demonstrate its utility in a different scenario?

Response. We thank the reviewer for this interesting suggestion, which has led to our substantially extended analyses presented in the paper. In defense of neuro-oncology: brain tumors represent the most common cause of cancer death in children, and effective therapies are lacking in both children and adults. The regulation of brain tumor plasticity is a central discussion in this big field. This said, we think it is a very interesting idea to explore scregclust's applicability to other cancer types. Accordingly, using the Curated Cancer Cell Atlas (3CA, <https://www.weizmann.ac.il/sites/3CA/>) as a starting point, we chose data sets representing 15 different cancer types (AML, breast, colorectal, glioblastoma, head and neck, liver, medulloblastoma, neuroblastoma, lung, osteosarcoma, ovarian, PDAC, prostate, renal and skin). For each of these data sets, we applied scregclust and merged the resulting output to a pan-cancer regulatory landscape (analogous to the analysis in Figure 4B). From this analysis, we found a total of 650 regulators out of which 122 were cancer-specific.

In addition, we scored each individual gene module against gene signatures for the 41 meta-programs from [10] and generated a regulator x meta-program matrix. Here the reader can inquire what regulator is predicted to most strongly interact with what meta-program. As an example, we find that the regulator most strongly associated with meta-program 12 (epithelial-to-mesenchymal transition) is PRRX1, a known regulator of EMT. This allows for generating new hypotheses of how to target specific meta-programs in cancer.

The pan-cancer analysis is described in Results p. 12-13, Figure 5, Supplementary Figure S7 and Supplementary Table 3.

4. ScRegClust involves several user-defined hyperparameters, such as lambda, the minimum size for non-empty clusters, and the initial value for K. Unreported and inconsistent hyperparameter values might lead to unstable performance and irreproducibility, as evidenced in Fig 2AB. Could the authors list all hyperparameters and their values for each analysis in this study? Furthermore, can they propose a strategy for selecting hyperparameters in typical use cases lacking ground truth,

even if it's as simple as default values? Does this strategy consistently yield satisfactory results against ground truth in a wide range of simulations, varying in data dimensions, module size, regulator count, and coefficient distribution? The supplementary material's simulations address some of these points but with limited variation in simulation parameters and focused only on lambda.

Response. Thank you for your suggestion. We agree that a detailed description of how hyperparameters are selected is crucial. All hyperparameters in each analysis in this study (where default values were not used) have now been listed in Supplementary Table 4. These include the target number of gene clusters (recommended range: 10-20), penalty parameter (recommended range: 0.01-0.05), and minimum cluster size (recommended value: 20). Throughout the manuscript, e.g. in the section "Regulatory-driven clustering: performance and robustness" (page 7-8) and in the Supplementary material, we present a number of simulation scenarios and two metrics (regulator importance and predictive R^2) which can guide the selection of the penalty parameter lambda. In the same section, we discuss how the number of starting clusters (K) can be chosen using a modified variant of the silhouette score, with several examples on simulated data provided in Figure S1. The minimal size of a nonempty cluster is less critical than the other two parameters and reflects the smallest size of a cluster (signature) that we are interested in.

To facilitate for the user in selecting hyperparameters, we have now included a section in the **Supplementary material called "Choosing hyperparameters" page 8-9**, intended to guide the typical user in which hyperparameters that are important to tailor in their analysis, how to do this and where in the manuscript they can find more information on these parameters and the selection strategy.

5. The current simulation methodology lacks detail. Could the authors mathematically describe their simulation methodology? Does it include a discrete sampling process and produce scRNA-seq read counts?

Response. Thank you for pointing this out. We agree that this could have been more carefully explained. We have now improved the presentation to describe how simulated data were generated (**Supplementary Material, Generation of simulated data, Page 3 and 4**). In brief, simulations were performed to investigate the statistical model and to demonstrate our procedure for model selection. As we describe in Supplementary Materials, scRNA-seq count data was used as a starting point, but after pre-processing, data was manipulated according to model assumptions to generate artificial datasets.

6. Fig S1B, 1DE: Employing a hypergeometric test could enhance result interpretation. Please report sample sizes, odds ratios, and P-values for each test. Also, clarify the definition of the background population in the Methods section.

Response. Thank you for the comment. We agree. We indeed used Fisher's test to show that the overlap in 1D is significant. Odds ratios and p-values are given (**see page 6 in Results**), as well as test details (**Methods page 16**). See also the response to Reviewers 1 and 2.

7. How are results integrated in Fig 4A?

Response. For the analysis in Figure 4 (and newly added Figure 5, described above) the algorithm is run on each dataset separately. Each run generates a “regulatory table”, a matrix of dimension regulators x modules. Thereafter, the regulatory tables from each run are combined by merging the matrices by row names (the function `merge(x,y,by='row.names', all=TRUE)` in R, to be specific). Finally, the merged regulatory table is centered and scaled and the generated z-scores are used for plotting. A clarification has been added to **Methods, page 16**.

8. The software can be installed. The tutorial can be partially run with errors quoted below. The outputs from the figures were not entirely error-free, and only a variant of Fig 2A was reproducible, albeit with discrepancies from the paper's version. The tutorial also resulted in over 70% of genes falling into the 'rag bag' cluster. The tutorial should represent a realistic use case and performance, and the authors need to clearly communicate anticipated outputs.

Errors encountered:

```
> plot_regulator_network(fit) Error in base::colSums(x, na.rm = na.rm, dims = dims, ...): 'x' must be an array of at least two dimensions
```

```
> for (i in 1:5){regulator_importance[,i] <- rowMeans(fit$results[[i]]$output[[1]]$importance, na.rm = TRUE) }
```

```
Error in regulator_importance[, i] <- rowMeans(fit$results[[i]]$output[[1]]$importance, : number of items to replace is not a multiple of replacement length
```

Response. We are grateful to the reviewer for taking the time to go through the tutorial and alert us to the error messages that arise. We realized that the tutorial on GitHub is not as up-to-date or well-commented as the vignette, which explains the error messages. To make the package more user-friendly, we have removed the old tutorial from GitHub and replaced it with a link to a website (scmethods.github.io/scregclust/) featuring the updated vignette and a thorough explanation of all functions included in the package. We hope that everything will run smoothly from now on.

Regarding the large number of genes in the rag-bag cluster, we would like to clarify that `scregclust` aims to find groups of genes that are sign-consistently regulated. If a gene ends up in the rag-bag cluster, then its correlation with the available regulator/sign-patterns associated with the available clusters is not strong enough to be included in one of the groups. Re-running `scregclust` on only the rag-bag genes to find additional clusters that might have been missed in the initial run has not proven fruitful, and genes are again sorted into the rag-bag cluster. This suggests that no sign-consistent regulator signatures can be found that regulate any substantial group of target genes in the rag-bag cluster. Since `scregclust` is neither a pure target gene clustering method nor focuses purely on regulatory link estimation, it is acceptable and desirable to only find groups of target genes that are sign-consistently regulated. Thus, 70% is a typical number, and this has been explained more fully in the documentation. WGCNA exhibits a similar behaviour in this regard as well.

That said, we acknowledge that it can occasionally be of interest to place target genes in one of the available clusters. In addition to the usual cluster label vector cluster, we have introduced an additional output vector cluster_all that places rag-bag genes in the cluster which explains it best (highest overall R^2 , despite it being under the set threshold).

Minor concerns: The use of $\sigma_{i,j}$ is confusing, as it seems independent of i .

Response. Thank you for the perceptive comment. $\sigma_{i,j}$ is truly allowed to vary with cluster i and target gene j such that different clusters are allowed to have different spread. It depends on the cluster through the chosen set of regulators R_i and on the target gene through the current **Response**. This can be seen in the sign-constrained linear model for target genes in Eq. 1 (equation numbers follow the revised manuscript): $Z_t^{(:,j)} = Z_r^{(:,R_i)} \text{diag}(s_i) B_i^{(:,j)} + \sigma_{i,j} e_{i,j}$. The reviewer is correct in stating that the plug-in estimate for residual variance in Eq. 4 truly is independent of cluster i due to how ordinary least squares for linear multi-response models works. This estimate is only used as a technical tool to stabilize the variance of target genes before coop Lasso estimation. However, the final estimate of $\sigma_{i,j}$ is performed with the help of the NNLS estimate in Eq. 7 and is dependent on both the cluster i and target gene j . A clarification of the statistical model and reasoning behind it has been added in **Methods, Page 17**.

For regulatory i not selected for cluster j , is it regarded as a target in cluster j ? This should be communicated and justified clearly in Methods.

Response. No, the assignment of genes to be tentative regulators or targets is provided at the start of the algorithm and not changed dynamically later. This has now been communicated and justified clearly in **Methods, Page 18**.

Fig 1C: What criteria were used by the authors to select specific regulators for visualization?

Response. Variance across modules.

Phrases like "Improving accuracy" as in "Regulatory-driven clustering improves accuracy" can be misleading. The authors should reword such claims to avoid confusion with improvements over existing methods. Additionally, any claims of improvement, even over initial conditions, should be substantiated through comprehensive comparisons.

Response. Thank you for the comment. The heading specifically referred to the observational fact that our method substantially improved clustering quality when compared to only clustering genes using kmeans++ in the simulation study described in the main manuscript, Figure 2 and in the Supplementary Material. We do agree that the phrasing can be made more exact and have changed that heading and carefully gone through the manuscript to avoid unnecessary jargon, subjective language, or comparative claims, consistent with journal style.

The manuscript alternates between "penalty parameter" and "penalization parameter" – consistency in terminology would be beneficial.

Response. Thank you for the comment. We have carefully revised the manuscript to ensure consistent terminology.

The paper made several mention of regulatory network, but scRegClust primarily reconstructs bipartite graphs, which lack several key features of typical regulatory networks, like perturbation propagation through chain reactions. The authors should clearly justify why scRegClust reconstructs regulatory networks, or consider other terms.

Response. Thank you for the comment. True, we are not estimating GRNs in the traditional perturbation biology sense, where genes are allowed to regulate each other. Rather we have developed a method that can give insight on how a predefined set of regulators affects a predefined set of target genes. We agree that this produces a bipartite GRN subgraph when regulators/targets of interest are fixed. We have not used the word bipartite graph in the text since this can be a bit abstract for many readers. The restrictions that the clustering and coop-lasso puts on the bipartite graph (target clusters sharing the same set of regulators and signs) is why scregclust is more stable than the network estimation methods.

References

- [1] Joseph M Rich, et al. *bioRxiv*, 2024.
- [2] X. Chen, et al. *Nat Commun*, 9(1):2949, Jul 2018.
- [3] C. Villalva, et al. *Int J Mol Sci*, 13(6):6983–6994, 2012.
- [4] I. Larsson, et al. *Mol Syst Biol*, 17(9):e10105, Sep 2021.
- [5] R. G. Verhaak, et al. *Cancer Cell*, 17(1):98–110, Jan 2010.
- [6] L. Garofano, et al. *Nat Cancer*, 2(2):141–156, Feb 2021.
- [7] E. Gangoso, et al. *Cell*, 184(9):2454–2470, Apr 2021.
- [8] L. Heumos, et al. *Nat Rev Genet*, 24(8):550–572, Aug 2023.
- [9] M. Crow, et al. *Genome Biol*, 17:101, May 2016.
- [10] A. Gavish, et al. *Nature*, 618(7965):598–606, Jun 2023.

REVIEWER COMMENTS

Reviewer #1 (Remarks to the Author):

The authors satisfactorily addressed most of our comments and we were able to validate known regulators in our own dataset. A couple of issues remain to be fixed.

1. Although the manuscript was revised to replace the term 'cluster' with 'module,' the R package (v1.8) still uses 'cluster.' Do the authors intend to update this in the future?
2. Row names on some of the heatmaps are crowded and difficult to read.

Reviewer #1 (Remarks on code availability):

The code works well with few issues and reproducible results. The authors also improved their tutorial. The new version shows improvements and runs with very minor issues. The package still uses a lot of memory which is understandable.

I tested it with multiple mixed population datasets and the results align with other analyses and biologically make sense.

Reviewer #2 (Remarks to the Author):

Reviewer #3 (Remarks to the Author):

The manuscript has been significantly improved in terms of clarity, details, and potential impact. However, as the details unveiled, several major concerns remained and became more apparent. It is increasingly clear that screglust relied on assumptions more suitable for bulk RNA-seq than scRNA-seq, used simulated data more similar to bulk RNA-seq for benchmarking, and compared against conventional methods widely applied on bulk RNA-seq. Lacking both simulation methods and competing methods for scRNA-seq data, its performance claims require additional evidence to justify. Without a comprehensive evaluation of its statistical performance in the context of existing methods, the impact of this study remains unclear. In particular, my following existing concerns remain:

1. While the authors provided the background of this study, the scope of screglust, and improved

benchmarking, I still have major concerns after reviewing their response.

a. While the benchmarking of traditional clustering algorithms is valuable, they may fail to account for the widely known challenges in scRNA-seq data. A google search of scRNA-seq biclustering or scRNA-seq gene clustering provided plenty of method papers, such as "An Effective Biclustering-Based Framework for Identifying Cell Subpopulations From scRNA-seq Data", "QUBIC2: a novel and robust biclustering algorithm for analyses and interpretation of large-scale RNA-Seq data", "scSTEM: clustering pseudotime ordered single-cell data", and "Celda: a Bayesian model to perform co-clustering of genes into modules and cells into subpopulations using single-cell RNA-seq data". These or similar studies appear to have sufficient relevance and importance to be referenced and benchmarked in this manuscript.

b. Similarly, more methods are proposed for gene regulatory network reconstruction from scRNA-seq, which can be challenging for traditional algorithms. However, very few are referenced and none is compared against.

c. To provide readers with sufficient background for existing research, the authors should perform a literature search and reference a few symbolic studies for each of the two fields above.

d. To put the performance of scregclust into proper context, the authors should include several methods from each of the two fields above for comprehensive benchmarking. Although scregclust is not expected to perform better than all other methods because it needs to balance two goals, putting its performance into a real context would allow better understanding and help readers to choose the method to use.

e. The authors' response provides a somewhat different intention of scregclust than the one presented in the abstract and introduction. For example, the current manuscript still describes scregclust as "a new method for the fast construction of regulatory programs". The manuscript should be updated accordingly in many places to reflect the actual intention described in this response.

5. The conclusions drawn from simulated data rest on the similarity of simulated data with real data. This study relies heavily on simulated data, particularly for benchmarking. However, without a discrete sampling process, simulated data would greatly differ from real scRNA-seq datasets. In fact, the normal distribution observed in this simulation is known to be violated in scRNA-seq data, leading to all the statistical challenges we face in recent years. The simulations for benchmarking need to include a discrete sampling step to generate count data to become sufficiently similar to real scRNA-seq data and lead to valid performance conclusions.

Response to comments by Reviewer #1 & 2

1. Although the manuscript was revised to replace the term 'cluster' with 'module,' the R package (v1.8) still uses 'cluster.' Do the authors intend to update this in the future?

Response: We are happy to hear that *scregclust* performed well on your in-house data sets. We have now updated the R-package to use the term module instead of cluster in the appropriate places.

2. Row names on some of the heatmaps are crowded and difficult to read.

Response: Thank you for pointing this out, we agree. We have adjusted the row names in figure 3B, 4B and 5A to make the figure more straightforward to read.

The code works well with few issues and reproducible results. The authors also improved their tutorial. The new version shows improvements and runs with very minor issues. The package still uses a lot of memory which is understandable.

I tested it with multiple mixed population datasets and the results align with other analyses and biologically make sense.

Response: Thank you for testing our package, we are glad to hear that it runs well.

Response to comments by Reviewer #3

1.a. While the benchmarking of traditional clustering algorithms is valuable, they may fail to account for the widely known challenges in scRNA-seq data. A google search of scRNA-seq biclustering or scRNA-seq gene clustering provided plenty of method papers, such as "An Effective Biclustering-Based Framework for Identifying Cell Subpopulations From scRNA-seq Data", "QUBIC2: a novel and robust biclustering algorithm for analyses and interpretation of large-scale RNA-Seq data", "scSTEM: clustering pseudotime ordered single-cell data", and "Celda: a Bayesian model to perform co-clustering of genes into modules and cells into subpopulations using single-cell RNA-seq data". These or similar studies appear to have sufficient relevance and importance to be referenced and benchmarked in this manuscript.

1b. Similarly, more methods are proposed for gene regulatory network reconstruction from scRNA-seq, which can be challenging for traditional algorithms. However, very few are referenced and none is compared against.

1c. To provide readers with sufficient background for existing research, the authors should perform a literature search and reference a few symbolic studies for each of the two fields above.

1d. To put the performance of scregclust into proper context, the authors should

include several methods from each of the two fields above for comprehensive benchmarking. Although scregclust is not expected to perform better than all other methods because it needs to balance two goals, putting its performance into a real context would allow better understanding and help readers to choose the method to use.

1e. The authors' response provides a somewhat different intention of scregclust than the one presented in the abstract and introduction. For example, the current manuscript still describes scregclust as "a new method for the fast construction of regulatory programs". The manuscript should be updated accordingly in many places to reflect the actual intention described in this response.

Response: Thank you for these valuable comments. We agree that it is important to consider software developed with an eye on single-cell data, and to contextualize our method further.

- Response to 1a (additional single cell clustering methods).
We agree that benchmarking against clustering methodology specifically designed for single-cell data is essential. To address this, we have expanded our analysis by including Celda [1] in our cluster stability evaluation, alongside k-means, hclust, and scregclust. Celda shows performance comparable to k-means and sometimes exceeds scregclust in terms of stability for smaller clusters, though scregclust demonstrates more consistent behavior overall. We also attempted to incorporate QUBIC2 into our benchmarking. However, due to the non-exclusivity of gene cluster membership, QUBIC2 was not suitable for our purposes. The majority of genes were not assigned to any cluster, and those that were assigned appeared in multiple clusters (See figures below). This issue, as noted in the Celda publication, is common in biclustering-based methods. The other methods you mentioned either faced reproducibility issues (DivBiClust) or were not applicable to our data analysis setting (scSTEM). Overall, scregclust demonstrates robust gene clustering performance, on par with or better than traditional and new single-cell clustering methods. We have cited these methods for readers who may seek alternative approaches.
- Response to 1b (additional network methods).
We agree and have extended our comparison to include additional methods. Our initial focus on GRNboost2 was due to its strong performance in a recent systematic evaluation of single-cell network construction methods (Pratapa et al., Nature Methods 2020). This study identified PIDC, GENIE3, and GRNBoost2 as the leading methods in terms of accuracy. We initially excluded GENIE3 due to its mathematical similarity to GRNBoost2, which is significantly faster. To broaden our analysis, we have now added PIDC and a single-cell adaptation of WGCNA (hdWGCNA) [2] to our supplementary figures (S2-S7). Scregclust performed exceptionally well in comparison to these methods, particularly in regulator stability and sign consistency.
- 1c. Literature Citations.
We agree that it's important to acknowledge advancements in clustering and

gene-to-gene network construction. We have added the relevant references to give proper credit to these developments.

- **1d. Comprehensive Benchmarking:**

This concern has been addressed through the additional benchmarking described above.

- **1e. Clarification of Scregclust's Purpose.**

We have further clarified in the Results section that the primary goal of scregclust is to detect clusters assigned to actionable regulators rather than constructing a complete gene-to-gene network. We believe that the term "regulatory program" effectively communicates this concept and is more intuitive than alternatives like "module networks," "gene batteries," or "bipartite graph." Additionally, we consider it a strength that our computational method development is conducted within the context of cancer research.

Figure (related to 1a): QUBIC2 applied to the Wang et al dataset. Clustered genes ordered by the number of clusters to which that gene was assigned.

Figure (related to 1a): QUBIC2 applied to the PBMC dataset. Clustered genes ordered by the number of clusters to which that gene was assigned.

2. The conclusions drawn from simulated data rest on the similarity of simulated data with real data. This study relies heavily on simulated data, particularly for benchmarking. However, without a discrete sampling process, simulated data would greatly differ from real scRNA-seq datasets. In fact, the normal distribution observed in this simulation is known to be violated in scRNA-seq data, leading to all the statistical challenges we face in recent years. The simulations for benchmarking need to include a discrete sampling step to generate count data to become sufficiently similar to real scRNA-seq data and lead to valid performance conclusions.

Response: It is correct that our previous submission, in Figure 2, used simulated data to illustrate the behavior of *scregclust*. Crucially, other than that, all benchmarking was performed on real data sets (count data). That said, we agree that our simulation approach was simplistic and have now updated Figure 2 so that samples are generated from a negative binomial distribution (i.e. counts) to more closely reflect single-cell data. The new simulation produces very similar results. We also clarify in the results part that the purpose of the simulation is to illustrate how to select appropriate penalty parameters for *scregclust*.

References:

- [1] <https://academic.oup.com/nargab/article/4/3/lqac066/6696781>
- [2] [https://www.cell.com/cell-reports-methods/fulltext/S2667-2375\(23\)00127-3](https://www.cell.com/cell-reports-methods/fulltext/S2667-2375(23)00127-3)

REVIEWERS' COMMENTS

Reviewer #3 (Remarks to the Author):

The authors addressed most of my concerns. In principle, more recent methods proposed for gene regulatory network reconstruction from scRNA-seq should be included for benchmarking and the ROC should be included in benchmarking. However, I also recognize the effort from the authors and would like to defer the judgement of method performance to the community.

As such, I recommend the publication of this manuscript and wish to congratulate the authors on this achievement!

Response to comment from Reviewer #3

1. *The authors addressed most of my concerns. In principle, more recent methods proposed for gene regulatory network reconstruction from scRNA-seq should be included for benchmarking and the ROC should be included in benchmarking. However, I also recognize the effort from the authors and would like to defer the judgement of method performance to the community.*

As such, I recommend the publication of this manuscript and wish to congratulate the authors on this achievement!

Response: We thank the reviewer for their kind feedback and for assessing our manuscript. We aimed to demonstrate the stability and reliability of *scregclust* in comparison to state-of-the-art clustering and network construction algorithms. Since our benchmarking emphasizes solution stability rather than the reconstruction of full gene-to-gene regulatory networks, we chose not to include a ROC curve in the analysis. We agree that users will benefit from selecting tools based on their specific goals, whether that is focusing on regulatory programs with *scregclust* or reconstructing full gene-to-gene networks.

Thank you again for your thoughtful review and for recommending publication.